# Belief-Calibrated Multi-Agent Consensus Seeking for Complex NLP Tasks

**Wentao Deng**
Shandong University
wentao.deng@mail.sdu.edu.cn

**Jiahuan Pei**
Vrije Universiteit
Amsterdam
j.pei2@vu.nl

**Zhiwei Xu**
Shandong University
zhiwei_xu@sdu.edu.cn

**Zhaochun Ren**
Leiden University
z.ren@liacs.leidenuniv.nl

**Zhumin Chen**[*]
Shandong University
chenzhumin@sdu.edu.cn

**Pengjie Ren**[*]
Shandong University
renpengjie@sdu.edu.cn

## Abstract

A multi-agent system (MAS) enhances its capacity to solve complex natural language processing (NLP) tasks through collaboration among multiple agents, where consensus-seeking serves as a fundamental mechanism. However, existing consensus-seeking approaches typically rely on voting mechanisms to judge consensus, overlooking contradictions in system-internal beliefs that destabilize the consensus. Moreover, these methods often involve agents updating their results through indiscriminate collaboration with every other agent. Such uniform interaction fails to identify the optimal collaborators for each agent, hindering the emergence of a stable consensus. To address these challenges, we provide a theoretical framework for selecting optimal collaborators that maximize consensus stability. Based on the theorems, we propose the **B**elief-**C**alibrated **C**onsensus **S**eeking (**BCCS**) framework to facilitate stable consensus via selecting optimal collaborators and calibrating the consensus judgment by system-internal beliefs. Experimental results on the MATH and MMLU benchmark datasets demonstrate that the proposed BCCS framework outperforms the best existing results by 2.23% and 3.95% of accuracy on challenging tasks, respectively. Our code and data are available at https://github.com/dengwentao99/BCCS.

## 1 Introduction

With the rapid advancement of large language models (LLMs), reasoning capabilities have become critical for tackling complex natural language processing (NLP) tasks. In multi-agent system (MAS), consensus-seeking has emerged as an essential protocol for enhancing collective reasoning through consensus evaluation and cooperative decision-making among agents [1]. Each agent may express its *opinion* by forming distinct or overlapping stances and judgments on a given task. Existing consensus-seeking approaches typically assess consensus by measuring the degree of agreement among agents [2, 3], and agents update their views by aggregating the opinions received from others, as illustrated in Figure 1(a). However, achieving robust collaboration in MAS consensus remains challenging: (1) current methods often overlook the underlying beliefs of individual agents when determining consensus, which may result in latent internal inconsistencies and compromise the overall stability of the consensus [4]; (2) agents generally lack mechanisms to selectively identify optimal collaborators, instead indiscriminately incorporating all received opinions. For instance, excessive

---

[*]Corresponding authors

39th Conference on Neural Information Processing Systems (NeurIPS 2025).

reliance on supportive agents may expedite convergence but risk producing suboptimal outcomes [5], while being overwhelmed by conflicting perspectives can impede consensus formation [6].

In response to the above two challenges and to facilitate stable consensus in MAS, we propose the **B**elief-**C**alibrated **C**onsensus **S**eeking (**BCCS**) framework to optimize the consensus-seeking process, as illustrated in Figure 1(b). To improve the accuracy of consensus determination, we introduce an enhanced *consensus judgement* module that evaluates whether consensus has been achieved across the entire MAS. A key element often overlooked in existing methods is **belief** [7], the degree of confidence each agent has in its own opinion. Hence, our consensus judgement module not only considers the agents' outputs but also calibrates them based on the associated belief levels. It categorizes the system into one of three consensus states: *full consensus*, *partial consensus*, or *no consensus*. In the case of *full consensus*, collaboration terminates and the consensus is output. When *partial consensus* occurs, which indicates the coexistence

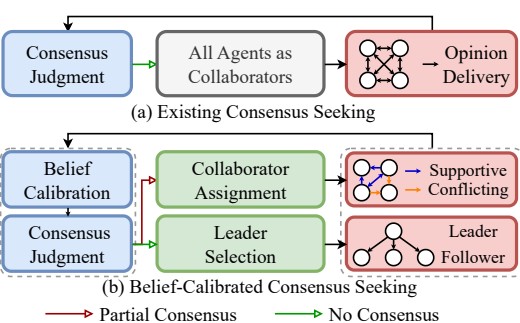

Figure 1: Comparison between previous consensus seeking methods and our proposed framework. (a) Existing consensus seeking methods. (b) Belief-Calibrated Consensus Seeking (BCCS).

of both supporting and opposing views, we invoke a *collaborator assignment* (CA) module that automatically assigns optimal collaborators to agents, thereby fostering convergence and avoiding suboptimal solutions. If the system enters a *no consensus* state with severe opinion divergence, the BCCS framework engages a *leader selection* (LS) module to identify and appoint leaders for each opinion group, guiding the direction of discourse and alleviating conflicts. Through repeated agent interactions and iterative updates of their viewpoints, a stable consensus can ultimately be reached and adopted as the final inference outcome. To ensure the theoretical soundness of BCCS, we formally establish the conditions under which stable consensus is guaranteed, specifically when (1) agents collaborate with both supportive and conflicting agents, and (2) agents follow leaders with diverse belief systems. This provides a theoretical foundation for the proposed BCCS framework.

In the experimental implementation, we evaluate the effectiveness of belief-calibrated consensus seeking (BCCS) on two widely-used benchmarks: MATH [8] and MMLU [9]. Results demonstrate that BCCS improves accuracy by 2.23% on MATH and 3.95% on MMLU compared to existing best results on challenge tasks. The main contributions of this study are summarized as follows:

- We propose the **B**elief-**C**alibrated **C**onsensus **S**eeking (**BCCS**) method to enhance the consensus-seeking process in multi-agent system (MAS). Specifically, BCCS incorporates a belief calibration mechanism where consensus judgement is calibrated based on agents' beliefs, and further integrates collaborator assignment and leader selection modules to promote consensus formation while mitigating suboptimal solutions.
- Theoretical guarantees are established for achieving stable consensus in two key scenarios: (i) cooperation involving both supportive and conflicting agents, and (ii) coordination among leaders with divergent beliefs. These theorems form the theoretical backbone of BCCS.
- Extensive experiments conducted on widely adopted benchmarks confirm the effectiveness of BCCS. Additionally, ablation studies are performed to quantify the impact of each core component.

## 2 Preliminaries

### 2.1 Multi-Agent Collaboration

**Task Formulation**  Consider a MAS comprising $n$ agents $A = \{a_1, \cdots, a_n\}$ and a user input question $q$. At the $k$-th round, each agent $a_i$ generates the opinion $o_i^k = (e_i^k, x_i^k)$, where $x_i^k$ is the answer to $q$, and $e_i^k$ represents the associated reasoning process. Following prior work [10], we adopt the output probability of LLMs as a proxy for belief. Specifically, the belief $b_i^k = P(x_i^k \mid q, e_i^k)$ of agent $a_i$ can be regarded as the generation probability, which estimates the determinacy of the model's output [7], where $P(\cdot)$ indicates the probability function. While LLM's output probabilities may not always perfectly reflect uncertainty, this approximation is a widely used and practical method for belief

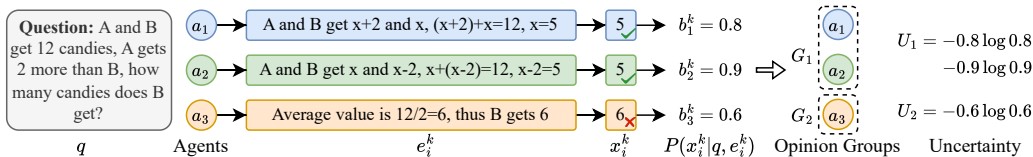

Figure 2: An illustration of the multi-agent system in NLP tasks, where each agent generates an answer $x_i^k$ along with its reasoning process $e_i^k$, the belief $b_i^k$ of $a_i$ is the generation probability.

estimation in large language models. When $a_i$ collaborates with other agents, it receives the opinions from the previous round to update its own opinion $o_i^{k+1} = a_i(q, \{o_j^k\}_{j=1}^n)$. Subsequently, the agents are clustered into $m$ opinion groups $G = \{G_p\}_{p=1}^m$, where each group $G_p$ contains a subset of agents grouped by topical similarity inferred from keyword distributions [11]. The uncertainty within $G_p$ is estimated by the information entropy [12] as $U_p = -\sum_{a_i \in G_p} b_i^k \cdot \log b_i^k$. Two groups are defined as *supportive* when their opinions converge and *conflicting* when they diverge. Each opinion group $G_p$ can have $n^l$ agents as leaders, guiding the opinion trends. After collaboration, the MAS selects the most frequently proposed answer as the final output. The illustration of multi-agent collaboration is shown in Figure 2. A summary of all notations used in this paper is provided in Table 14.

**Consensus Analysis**    We investigate the conditions for stable consensus in MAS through the lens of *opinion dynamics* [13]. For each agent $a_i \in G_p$, the answer and belief are updated according to $x_i^{k+1} = x_i^k + u_i^k$ and $b_i^{k+1} = b_i^k + v_i^k$, where $u_i^k$ and $v_i^k$ are the respective update increments. Since the agents often employ averaging strategies to update their answers [14], we define $u_i^k = \alpha \sum_{a_j \in A_i^*}(x_j^k - x_i^k)$, where $A_i^*$ refers to either the supportive agents $A_i^s$ or conflicting agents $A_i^c$ which indicate the collaborated agents from corresponding opinion groups of $G_p$. Due to the supportive agents tend to align the beliefs [15], the belief update is given by $v_i^k = \beta \sum_{a_j \in A_i^s}(b_j^k - b_i^k)$. In contrast, conflicting agents drive belief divergence [16], resulting in $v_i^k = -\beta \sum_{a_j \in A_i^c}(b_j^k - b_i^k)$. In this paper, we set the step sizes $\alpha$ and $\beta$ as $\frac{2}{n}$ as in [13].

## 2.2 Consensus Judgment

The current consensus is primarily judged by *Byzantine Consensus* [3], which ensures that the multi-agent system reaches consensus when more than $\frac{2}{3}$ of the agents reach the same conclusion. For the dominant consensus group $A^s$ which indicates the group of agents with the largest number of identical conclusion and other agents form the conflict group $A^c$, Byzantine Consensus can be represented as follows:

$$p_s^k = \frac{|A^s|}{|A^s| + |A^c|} > \frac{2}{3} \Rightarrow \frac{|A^s|}{|A^c|} > 2, \tag{1}$$

where $p_s^k$ indicates the proportion of the dominant consensus group. Therefore, existing methods only take the agents' answers as the basis for consensus judgment, neglecting the underlying beliefs that carry important implicit information.

## 3 Theoretical Foundations of Stable Consensus

To analyze the consensus of MAS, we follow the definition of global stability in learning dynamics [17] to formally define the stable consensus state. Then sufficient conditions for achieving such consensus are derived accordingly, with detailed proofs provided in Appendix B.

**Definition 3.1.** (**Stable Consensus**)    Given a MAS comprising $n$ agents $\{a_i\}_{i=1}^n$, and any initial answers $\{x_i^1\}_{i=1}^n$, the system is said to reach a stable consensus answer if the following two conditions are satisfied: (1) agents' answers converge to consensus, and (2) each agent's belief of the answer is coherent with the beliefs of other agents.

**Theorem 3.2.** *Let $\{x_i^k\}_{i=1}^n$ denote the opinions and $\{b_i^k\}_{i=1}^n$ denote the beliefs of a MAS with $n$ agents at the $k$-th step of collaboration. The collaboration between agents satisfies the following properties:*

1. *When each agent in MAS collaborates with supportive agents, the MAS tends to reach the stable consensus, converging to the state of the average opinion and belief of all collaborating agents.*
2. *When any agent in MAS collaborates with conflicting agents, the MAS tends to form the unstable consensus, potentially leading to divergence or oscillation in group states.*

**Theorem 3.3.** *Let $\{x_i^k\}_{i=1}^n$ and $\{b_i^k\}_{i=1}^n$ represent the opinions and beliefs of a MAS with $n$ agents at the $k$-th step. Within each opinion group, the $i$-th agent follows $n^l$ leaders, and the collaboration between followers and their respective leaders satisfies the following properties:*

1. *When each agent in an opinion group collaborates with its leaders, the MAS tends to reach the stable consensus, converging to the average state of the leaders.*
2. *When the leaders' average belief is higher than other agents' beliefs, the leaders with higher beliefs can expedite the convergence to the stable consensus.*

## 4 Methodology

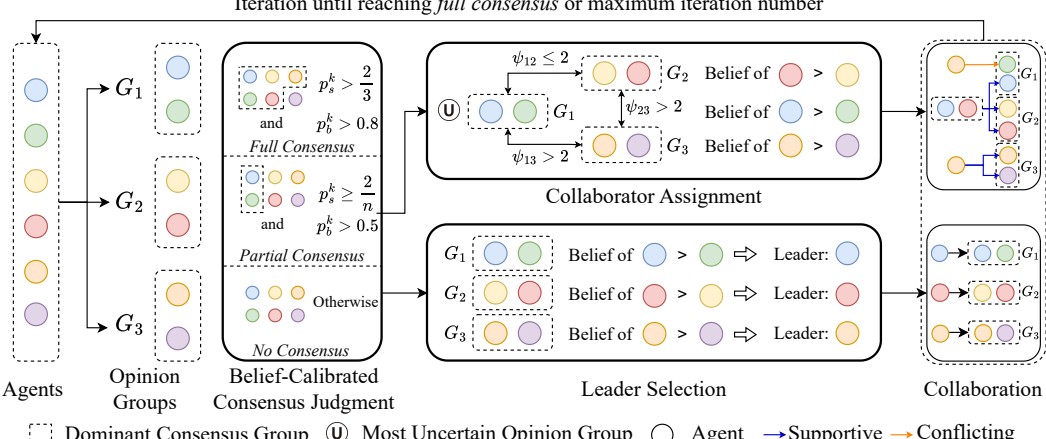

Figure 3: An illustration of the Belief-Calibrated Consensus Seeking (BCCS) framework. The arrows represent the workflows. After obtaining opinion groups, the belief-calibrated consensus judgment (BCCJ) module judges the consensus state of MAS. If MAS reaches *partial consensus*, the collaborator assignment (CA) module estimates the conflict levels between each two opinion groups through conflict scores, then assigns the collaborators for agents in each opinion group. If MAS reaches *no consensus*, the leader selection (LS) module selects leaders for each opinion group. The processes above iterate until reaching *full consensus* or maximum iteration number.

We propose the BCCS framework, grounded in the emerging paradigm of LLM-driven MAS [18]. The overall architecture is illustrated in Figure 3, which process is shown in Algorithm 1. Motivated by theoretical insights into the conditions required for achieving stable consensus, BCCS is designed to enhance consensus-seeking by iteratively executing three core modules: (1) The belief-calibrated consensus judgment (BCCJ) module judges the consensus status of the MAS based on agents' individual answers and beliefs, classifying it as one of three states: *full consensus*, *partial consensus*, or *no consensus*. (2) In the case of *partial consensus*, the collaborator assignment (CA) module assigns the most suitable collaborator to each agent to encourage convergence. (3) When *no consensus* is detected, the leader selection (LS) module designates the leaders for each opinion group to facilitate consensus building. The iteration terminates when the MAS reaches *full consensus* or the maximum number of iterations is exceeded. The final answer is determined as the most frequently agreed-upon conclusion among all agents. Detailed implementations of each module are described below.

### 4.1 Belief Calibrated Consensus Judgment

In the $k$-th round of collaboration, agents generate the collaborative outputs $\{x_i^k\}_{i=1}^n$ along with their corresponding beliefs $\{b_i^k\}_{i=1}^n$ in response to the input question $q$. Unlike prior studies that solely rely on agents' explicit answers to judge consensus, the BCCJ module also incorporates agents' beliefs for calibration. It categorizes the consensus state of the MAS into three levels as follows.

**Full Consensus** A *full consensus* state is declared when a substantial majority of agents reach consensus with high belief levels. Specifically, the proportion $p_s^k$ of agents in the dominant consensus group must exceed $\frac{2}{3}$, as described in Section 2.2. Additionally, to avoid convergence to a suboptimal consensus, the average beliefs within the dominant group must be at least twice that of the conflicting group, according to the condition analogous to Equation (1):

$$p_s^k > \frac{2}{3}, \qquad \frac{\sum\limits_{a_i \in A^s} b_i^k}{|A^s|} > 2 \frac{\sum\limits_{a_i \in A^c} b_i^k}{|A^c|} \Rightarrow \frac{\sum\limits_{a_i \in A^s} b_i^k}{\sum\limits_{a_i \in A^c} b_i^k} > 4 \Rightarrow p_b^k = \frac{\sum\limits_{a_i \in A^s} b_j^k}{\sum\limits_{a_i \in A^s} b_i^k + \sum\limits_{a_i \in A^c} b_i^k} > 0.8. \quad (2)$$

Thus, the MAS state is recognized as full consensus only if $p_s^k > \frac{2}{3}$ and $p_b^k > 0.8$.

**Partial Consensus** When a subset of agents reaches consensus with moderate beliefs, the system is considered to be in a *partial consensus* state. In this case, at least two agents must form a dominant group whose aggregate belief in the consensus surpasses that of the conflicting group. The condition is formally defined as:

$$p_s^k \geq \frac{2}{n}, \qquad \sum_{a_i \in A^s} b_i^k > \sum_{a_i \in A^c} b_i^k \Rightarrow p_b^k = \frac{\sum\limits_{a_i \in A^s} b_i^k}{\sum\limits_{a_i \in A^s} b_i^k + \sum\limits_{a_i \in A^c} b_i^k} > 0.5. \quad (3)$$

Therefore, when the preceding condition is satisfied but Equation (2) is not, the BCCJ module categorizes the state as partial consensus.

**No Consensus** A *no consensus* state is identified when neither the full nor partial consensus conditions (Equation (2) and (3)) are satisfied, indicating that no agents have reached agreement at a sufficiently high belief level.

The BCCS framework adopts distinct strategies for each of these states. Upon achieving full consensus, the collaboration terminates, and the resulting consensus is returned as the final output. Otherwise, the collaboration continues until full consensus is achieved or the predefined maximum number of interaction rounds is reached. For cases of partial or no consensus, two auxiliary modules are subsequently activated to facilitate consensus among agents.

## 4.2 Collaborator Assignment

In a partial consensus state, agents in the MAS may exhibit both supportive and conflicting relationships. Theorem 3.2 demonstrates that collaboration with supportive agents enables the MAS to reach a stable consensus, whereas collaboration with conflicting agents leads to an unstable one. However, relying solely on supportive agents may result in suboptimal solutions [5]. Therefore, agents with uncertain opinions must interact with both supportive and conflicting agents to ensure balanced decision-making. Within the BCCS framework, the most uncertain opinion group is identified as $G_u = \arg\max_{G_p} \{U_p\}_{p=1}^m$. To prevent *least reliable agent* with lowest belief in $G_u$ from continuously driving the collaboration toward suboptimal solutions, the CA module selects the agent with the highest belief from the corresponding *conflicting group* to guide the opinion change. Conversely, to ensure all other agents that are *more reliable* continue to guide the process toward optimal consensus, the module selects the agents with the highest belief from the corresponding *supportive groups* to collaborate. By tailoring collaboration strategies based on the belief levels of agents, the CA module promotes convergence under *partial consensus* while effectively avoiding suboptimal outcomes.

To quantify the degree of conflict between opinion groups $G_p$ and $G_q$, and to identify which groups are supportive and which are conflicting with respect to a given group, we propose a *conflict score* $\psi_{pq}$ that captures both macro- and micro-level perspectives. It comprises the following two components.

**Macro-Conflict** The macro-conflict estimates the overall conflict level between $G_p$ and $G_q$ by measuring the proportion of belief inconsistencies among all opinions in both groups. The calculation of macro-conflict $\psi_{pq}^{\mathcal{G}}$ is shown in Equation (5), where $G_p \oplus G_q$ represents the complementary set of agents with the same answers. Notably, $\psi_{pq}^{\mathcal{G}}$ denotes the belief-weighted complement of the Jaccard similarity [19]. Following the Jaccard similarity threshold [19], $G_p$ and $G_q$ are considered to be in macro-conflict when $\psi_{pq}^{\mathcal{G}} \geq 0.5$.

**Micro-Conflict** The micro-conflict estimates the difference of local consistency between $G_p$ and $G_q$. According to Equation (2), we define the local consistency score as $\Theta_* = \sum_{a_i \in G_*^s} b_i^k - 4 \sum_{a_i \in G_*^c} b_i^k$

and $\Theta_* > 0$ is the condition of full consensus. $G_*^s$ and $G_*^c$ indicate the opinion groups of agents with the agreed answers and disagreed answers, where $*$ indicates $p$ or $q$. The difference of local consistency is estimated by the distance between the local consistency scores of $G_p$ and $G_q$, and the lower bound is given in Equation (4). When the lower bound is strictly positive, it indicates a significant consensus difference between $G_p$ and $G_q$, which implies a micro-conflict when $\psi_{pq}^{\mathcal{L}} > 4$ in Equation (5).

$$|\Theta_p - \Theta_q| \geq \left| \sum_{a_i \in G_p^s} b_i^k - \sum_{a_i \in G_q^s} b_i^k \right| - 4 \left| \sum_{a_i \in G_p^c} b_i^k - \sum_{a_i \in G_q^c} b_i^k \right|. \tag{4}$$

$$\psi_{pq}^{\mathcal{G}} = \frac{\sum\limits_{a_i \in G_p \oplus G_q} b_i^k}{\sum\limits_{a_i \in G_p \cup G_q} b_i^k}, \qquad \psi_{pq}^{\mathcal{L}} = \frac{\left| \sum\limits_{a_i \in G_p^s} b_i^k - \sum\limits_{a_i \in G_q^s} b_i^k \right|}{\left| \sum\limits_{a_i \in G_p^c} b_i^k - \sum\limits_{a_i \in G_q^c} b_i^k \right|}, \qquad \psi_{pq} = \psi_{pq}^{\mathcal{G}} \cdot \psi_{pq}^{\mathcal{L}}. \tag{5}$$

The conflicting score $\psi_{pq}$ incorporates both macro- and micro-conflicts. Accordingly, $G_p$ and $G_q$ are considered to be in conflict when $\psi_{pq} > 2$. Besides, one opinion group is always self-supporting.

### 4.3 Leader Selection

When the multi-agent system reaches the no consensus state, there are no mutually supportive opinion groups in the system. According to the Theorem 3.3, it follows that selecting the agent with the highest belief value from each opinion group as a leader enables the opinions to converge most rapidly to the average of the leaders' beliefs. Accordingly, the LS module selects the $n^l$ agents with the highest belief in each group, denoted as $A^l$, to serve as leaders, while the remaining agents update their opinions by interacting exclusively with these leaders.

It can be found that selecting leaders with lower beliefs compromises the system's robustness by reducing consensus reliability, while higher-belief leaders facilitate faster convergence. Our proposed method selects the agent with the highest beliefs as the leaders in each iteration to avoid non-robust outcomes, thereby preventing suboptimal agents from serving as long-term leaders. Situations where all agents have relatively low belief are rare. If such a case does occur, it indicates that none of the agents are capable of solving the problem, making it impossible to accomplish the task through collaboration mechanism, instead it will complete the task through the voting mechanism.

## 5 Experiments

In the experiments, we seek to answer the following research questions:

- **RQ1:** How does the performance of BCCS compare to existing single-agent and multi-agent methods in NLP tasks?
- **RQ2:** How does each functional component of BCCS contributes to the performance?
- **RQ3:** How do the supportive/conflicting agents and leaders impact the consensus in BCCS?

### 5.1 Baselines and Benchmarks

To validate the effectiveness of BCCS, we evaluate the single-agent and multi-agent methods for comparsion. The single-agent methods include CoT [20], Reflection [21], CoT-SC [22]. The mult-agent methods include EoT [23], GroupDebate [24], MAD [18], PARSE [25], DyLAN [3] and CMD [26]. All experiments are conducted on two NLP benchmark datasets, including MATH [8] with 7 types of mathematical reasoning problems, and MMLU [9] with 4 primary types of natural language understanding tasks. Further details on the baselines and datasets can be found in Appendix C.1 and C.2. The performance of all methods is evaluated in terms of accuracy.

### 5.2 Implementation Details

For each datasets, we randomly sample three groups of 500 examples with random seeds 100, 200, and 300 to conduct three independent experiments. The final results present the mean performance across

runs, along with the corresponding standard errors of mean (SEM) [27] shown as error bars. More implementation details can be found in Appendix C.4. In the main experiments, the determination for optimal number of agents and iteration rounds is consistent with common practices in the existing multi-agent collaboration systems [18], where such hyperparameters are often set empirically. Specifically, the number of agents is $n = 7$ and the maximum iteration rounds is 3 across all methods. The number of leaders is set as $n^l = 2$. A detailed ablation study of these hyperparameters is available in Appendix D.2. Unless stated otherwise, each agent employs Qwen2.5-7B-Instruct as the backbone model.

Table 1: Main results on the MATH dataset. Bold numbers indicate the best-performing results among all methods.

| Method | Algebra | Counting & Probability | Geometry | Intermediate Algebra | Number Theory | Prealgebra | Precalculus | #Avg |
|---|---|---|---|---|---|---|---|---|
| CoT | 91.64±0.56 | 74.30±4.55 | 58.98±5.46 | 52.61±2.91 | 71.33±4.34 | 85.53±1.71 | 57.59±3.94 | 73.33±1.07 |
| Reflection | 91.83±1.88 | 76.98±1.98 | 61.55±3.85 | 52.58±2.33 | 72.57±0.29 | 87.65±1.26 | 59.89±5.73 | 74.67±0.81 |
| CoT-SC | 92.15±1.12 | 73.91±0.60 | 61.76±7.00 | 62.87±0.73 | 74.93±4.30 | 85.52±1.70 | 63.93±5.58 | 76.67±0.18 |
| EoT | 94.85±1.27 | 77.87±4.31 | 63.03±6.43 | 60.75±1.21 | 80.74±1.78 | 89.42±0.91 | 61.38±6.81 | 78.40±0.31 |
| GroupDebate | 94.07±1.35 | 78.37±2.73 | 67.70±6.51 | 59.98±1.62 | 75.33±3.81 | 89.08±0.94 | 61.89±5.35 | 77.93±0.84 |
| MAD | 94.05±0.39 | 78.37±1.76 | 66.14±7.16 | 62.09±1.99 | 79.57±1.36 | 90.15±0.81 | 62.01±3.68 | 78.87±0.18 |
| PARSE | 94.84±0.83 | 76.88±1.04 | 68.31±5.51 | 61.13±3.00 | 80.85±0.29 | 88.76±0.93 | 59.14±3.76 | 78.53±0.55 |
| CMD | 95.11±0.92 | 75.59±2.94 | 67.81±7.22 | 61.17±1.75 | 81.65±2.37 | 90.16±0.39 | 61.21±4.25 | 78.93±0.53 |
| DyLAN | 95.15±0.81 | 76.29±2.95 | 67.08±7.90 | 59.94±2.03 | 80.74±1.78 | 90.09±1.71 | 62.70±5.19 | 78.80±0.31 |
| BCCS | **95.41**±0.76 | **79.07**±1.12 | **68.64**±7.39 | **64.28**±1.60 | **82.81**±1.74 | **90.88**±0.14 | **64.93**±5.17 | **80.60**±0.23 |

Table 2: Main results on the MMLU dataset.

| Method | STEM | Social Sciences | Humanities | Other | #Avg |
|---|---|---|---|---|---|
| CoT | 68.70±1.24 | 78.19±0.82 | 71.84±1.25 | 70.50±2.95 | 71.87±0.96 |
| Reflection | 70.93±1.94 | 78.81±1.56 | 72.99±1.52 | 70.79±1.84 | 73.07±1.67 |
| CoT-SC | 72.76±0.73 | 78.82±1.12 | 71.84±2.24 | 69.61±3.00 | 73.13±1.33 |
| EoT | 75.81±0.54 | 76.01±1.89 | 73.56±2.07 | 71.39±2.95 | 74.33±1.48 |
| GroupDebate | 77.03±0.81 | 78.50±1.08 | 71.26±2.74 | 71.98±2.81 | 74.87±1.54 |
| MAD | 78.46±1.66 | 78.50±1.62 | 73.85±2.01 | 72.86±1.80 | 76.13±1.46 |
| PARSE | 78.05±1.27 | 79.44±1.43 | 74.14±1.99 | 73.74±1.56 | 76.47±0.48 |
| CMD | 76.63±1.02 | 78.82±1.12 | 72.41±2.28 | 71.98±2.36 | 75.07±1.44 |
| DyLAN | 78.25±0.89 | 77.26±2.43 | 74.21±2.23 | 69.03±1.84 | 75.00±1.51 |
| BCCS | **79.47**±0.81 | **80.69**±1.65 | **78.16**±3.20 | **75.22**±2.66 | **78.47**±1.22 |

# 6 Results and Discussions

## 6.1 Overall Performance

To address **RQ1**, we compare BCCS with several baselines on MATH and MMLU. The results are reported in Table 1 and Table 2. BCCS outperforms the baselines consistently in both datasets. Specifically, BCCS outperforms the strongest multi-agent methods by 1.67%/2.00% in terms of average accuracy on MATH/MMLU. The improvements are more significant on more challenging tasks, with a maximum increase of 2.23% and 3.95% on MATH and MMLU respectively. This is because BCCJ can ensure a sufficiently high belief level in the consensus results, and CA and LS can select the optimal agents as collaborators and leaders, which facilitate the MAS reaching stable consensus and avoid the suboptimal results. Besides, the results in more scenarios of NLP tasks are shown in Section D.1. For simple tasks, since the original model itself can already achieve relatively stable consensus, the marginal benefit of further introducing collaborative mechanisms is relatively limited.

Besides, the multi-agent methods are more effective than single-agent methods. This is because in multi-agent collaboration, agents can refine their own answers by incorporating opinions from others, making the system resilient to errors from individual agents.

Table 3: Ablation study on the MATH dataset. Bold numbers indicate the best-performing results among all conditions. "-" indicates removing the corresponding module, and "-Conflict" indicates using supportive opinions only. "R.Leader" indicates selecting $n^l$ leaders randomly.

| Method | Algebra | Counting & Probability | Geometry | Intermediate Algebra | Number Theory | Prealgebra | Precalculus | #Avg |
|--------|---------|------------------------|----------|----------------------|---------------|------------|-------------|------|
| BCCS | **95.41**±0.76 | **79.07**±1.12 | **68.64**±7.39 | **64.28**±1.60 | **82.81**±1.74 | **90.88**±0.14 | **64.93**±5.17 | **80.60**±0.23 |
| -CA | 95.10±1.34 | 76.89±1.99 | 66.25±7.79 | 60.02±1.39 | 79.18±2.42 | 90.13±0.92 | 62.27±4.55 | 78.60±0.12 |
| -Conflict | 95.13±0.53 | 76.39±3.47 | 63.80±7.12 | 61.63±1.61 | 82.68±2.08 | 89.03±1.74 | 60.68±5.25 | 79.00±0.23 |
| -LS | 94.05±0.39 | 76.39±4.22 | 66.14±7.16 | 60.47±3.67 | 79.45±1.58 | 88.74±0.62 | 62.49±5.47 | 78.13±0.48 |
| R.Leader | 95.14±0.11 | 76.98±2.32 | 63.53±7.78 | 59.99±0.43 | 81.64±1.45 | 89.44±0.85 | 60.80±7.13 | 78.33±0.66 |
| -BCCJ | 94.86±0.38 | 76.09±0.60 | 64.42±6.24 | 58.49±2.20 | 79.18±2.42 | 89.84±0.51 | 62.21±3.54 | 78.00±0.53 |

Table 4: Ablation study on the MMLU dataset.

| Method | STEM | Social Sciences | Humanities | Other | #Avg |
|--------|------|-----------------|------------|-------|------|
| BCCS | **79.47**±0.81 | **80.69**±1.65 | **78.16**±3.20 | **75.22**±2.66 | **78.47**±1.22 |
| -CA | 78.25±1.24 | 79.13±0.82 | 75.28±2.87 | 73.45±1.84 | 76.67±0.29 |
| -Conflict | 78.66±1.61 | 80.37±0.54 | 76.43±3.24 | 72.86±2.57 | 77.20±0.81 |
| -LS | 77.03±1.33 | 79.44±0.93 | 77.87±2.35 | 72.57±3.11 | 76.73±1.20 |
| R.Leader | 79.27±1.06 | 79.13±0.82 | 73.85±3.16 | 73.16±3.40 | 76.60±1.06 |
| -BCCJ | 77.44±1.54 | 79.75±1.36 | 77.30±2.74 | 74.04±3.87 | 77.13±1.01 |

## 6.2 Ablation Study

To address **RQ2**, we ablate the functional modules to evaluate their impact on the performance, including belief-calibrated consensus judgment (BCCJ) (w.r.t "-BCCJ"), collaborator assignment (CA) (w.r.t "-CA") and leader selection (LS) (w.r.t "-LS"). Specifically, "-BCCJ" replaces BCCJ with *Byzantine Consensus* [3]. Besides, to evaluate the impact of conflicting agents and leaders with highest beliefs, we also conduct experiments excluding conflicting agents (w.r.t "-Conflict") and randomly selected leaders (w.r.t "R.Leader"). The results are reported in Table 3 and Table 4.

All modules of BCCS have positive influence on performance on the two datasets. After removing the modules of BCCJ, CA, and LS, the values of average accuracy decrease 2.60%/1.34%, 2.00%/1.80% and 2.47%/1.74% on the MATH and MMLU benchmarks, respectively. These results indicate that the modules of BCCJ, CA, and LS can facilitate the MAS reaches a consensus with the correct answer. Moreover, both "-Conflict" and "R.Leader" exhibit performance decrease. This is because collaborating solely with supportive agents can lead to suboptimal solutions and lead to incorrect answers, while leaders with lower beliefs may cause other agents to converge on unreliable opinions, ultimately reducing the system's decision quality.

## 6.3 Analysis Experiments

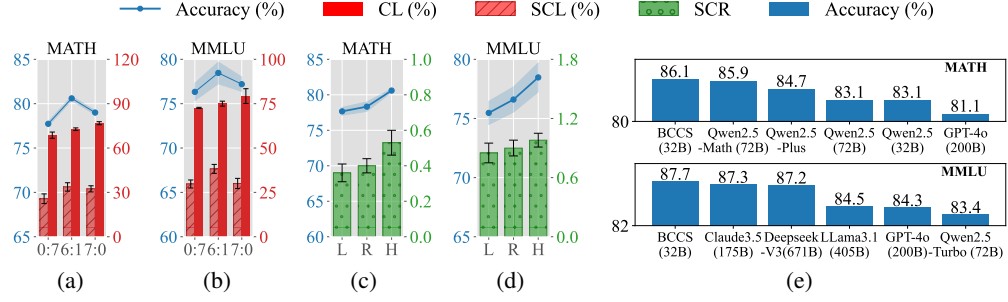

Figure 4: (a) and (b): the results of the $CL$, $SCL$ and accuracy for each supportive-to-conflicting collaboration ratio (including "0:7", "6:1" and "7:0"). (c) and (d): the results of the $SCR$ and accuracy for the lowest, random and highest leaders' beliefs (denoted as "L", "R", "H"). (e): the analysis of the parameter size, the x-axis denotes the LLMs for comparison.

To address **RQ3**, we analyze the impact of supportive-to-conflicting collaboration ratio and the leader belief on consensus and performance, with case studies in Appendix D.7. Inspired by agreement level [2], we calculate the (success) consensus level $(S)CL$ as the average ratio of (success) consensus agents per case, and the success consensus rate $SCR$ as the average number of success consensus agents per round per case, which calculation are represented in Equation (6). $n_{case}$, $n$, $n_u^s$, $n_u^r$ indicate the numbers of cases, agents, consensus agents in $u$-th case and rounds in $u$-th case, $x_u$ and $x_u^*$ indicate the consensus answer and ground-truth of $u$-th case. $\mathbf{1}[\cdot]$ is indicator function.

$$CL = \frac{\sum\limits_{u=1}^{n_{case}} \frac{n_u^s}{n}}{n_{case}}, \qquad SCL = \frac{\sum\limits_{u=1}^{n_{case}} \frac{n_u^s \cdot \mathbf{1}[x_u = x_u^*]}{n}}{n_{case}}, \qquad SCR = \frac{\sum\limits_{u=1}^{n_{case}} \frac{n_u^s \cdot \mathbf{1}[x_u = x_u^*]}{n_u^r}}{n_{case}}. \tag{6}$$

**Effect of supportive/conflicting collaboration for consensus**   We demonstrate the impact of the supportive-to-conflicting collaboration ratios (we set "0:7", "6:1", "7:0") on $CL$, $SCL$ and accuracy in Figure 4 (a) and (b). As the supportive collaboration ratio rises, the consensus level ($CL$) increases. Collaborating with the supportive agents can facilitate the MAS reaching stable consensus as shown in Theorem 3.2. The more agents collaborate with these supportive agents, the greater the consensus achieved. Besides, as $CL$ increases, the success consensus level ($SCL$) and accuracy demonstrate a trend of increasing first and then decreasing. This is because collaborating solely with supportive agents lead to suboptimal solutions, consequently causing a decline in $SCL$ and accuracy.

**Effect of leaders with different beliefs for consensus**   We demonstrate the impact of the leaders' beliefs in Figure 4 (c) and (d). We analyze and compare the performance of BCCS of lowest, random and highest leaders' beliefs (denoted as "L", "R", "H") on $SCR$ and accuracy. As leaders' beliefs increase, both the success consensus rate ($SCR$) and accuracy improve. Higher-belief leaders expedite stable consensus convergence when their average belief exceeds other agents', as shown in Theorem 3.3, whereas those with lower average belief may reduce consensus reliability.

**Analysis of Model Sizes**   To compare BCCS with different sized models on full datasets, we use Qwen2.5-32B-Instruct as backbone model on MATH, and DeepSeek-R1-Distill-Qwen-32B on MMLU. The results are shown in Figure 4 (e). BCCS outperforms same-scale models and matches or surpasses larger-scale models, which demonstrate consistent scalability across varying model sizes, with performance benefits persisting even at larger parameter sizes. Besides, we report the comparison results between BCCS and state-of-the-art baselines with different model sizes in Appendix D.3.

## 7   Related Works

### 7.1   Multi-Agent Collaboration

The multi-agent collaboration [18, 28] refers to a system where multiple agents collaborate to achieve a common goal, the dominant work focuses on updating the opinion of each agent based on those opinions from other agents to reach the consensus result [29–33]. CMD [26] and EoT [23] share opinions between agents and update results. SPP [34] and DyLAN [3] assume agents as distinct roles to address different aspects, and integrate solutions for the final answer. The debate based methods [35–37] adopt multiple agents engage in debates to solve problems. MADKE [38], MAD [18], Heter-MAD [39], and GroupDebate [24] output results by reaching consensus or summarizing opinions by another agent. PARSE [25] permanently assigns collaborators based on predefined collaboration structure. BENCHFORM [40] explores the conformity in the multi-agent systems. However, these methods can not identify the optimal collaborators for each agent, hindering the emergence of a stable consensus. Different from existing methods, we establish the theoretical foundation for selecting collaborators and leaders which can facilitate stable consensus in MAS, and propose the CA and LS modules for selecting collaborators and leaders to facilitate stable consensus.

### 7.2   Multi-Agent Consensus Seeking

The multi-agent consensus seeking [14, 41] aims to facilitate the MAS's consensus. Traditional consensus seeking methods adopt the consensus protocol to facilitate and determine consensus [13, 42, 43] and adjust game strategies through belief [44, 45]. Existing LLM-based consensus seeking methods rely on voting mechanisms for consensus judgment. AAD [2] outputs the consensus results

via the majority voting [46, 47]. Byzantine consensus theorem [48–50] are widely used to determine whether the MAS reaches consensus exceeds $\frac{2}{3}$ [3, 51]. However, these methods overlook the contradictions in system-internal beliefs that destabilize the consensus. Different from existing works, we propose the BCCJ module to calibrate the consensus judgment by system-internal beliefs.

# 8 Conclusion and Future Work

In this paper, we provide a theoretical framework for selecting optimal collaborators that maximum consensus stability. Based on the theorems, we propose the belief-calibrated consensus seeking (BCCS) framework to facilitate stable consensus via selecting optimal collaborators and calibrating the consensus judgment by system-internal beliefs. The experimental results confirm the effectiveness of BCCS and demonstrate that BCCS can facilitate the MAS reaching consensus and avoid falling into suboptimal solutions. In future work, we will explore the dynamic leader selection for enhancement.

**Broader Impacts** Our method requires no extra training or data, ensuring ease of use. While current multi-agent systems face scalability limits and risks like harmful LLM behavior, BCCS addresses this by selecting optimal agents as collaborators to ensure stable consensus.

# 9 Limitations

Although BCCS performs well in natural language processing tasks, we have not evaluated it in an embodied intelligence environment.

# Acknowledgements

We would like to thank the editors and reviewers for their helpful comments. This research was supported by the National Key R&D Program of China with grant No. 2024YFC3307303, the Natural Science Foundation of China (62472261, 62372275), the Technology Innovation Guidance Program of Shandong Province with grant No. YDZX2024088, the Provincial Key R&D Program of Shandong Province with grant No. 2024CXGC010108.

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

# A   Algorithm

---
**Algorithm 1** Algorithm of belief-calibrated consensus seeking (BCCS)
---
**repeat**
   Clustering opinions via KMeans and obtain opinion groups
   **Belief-calibrated consensus judgment (BCCJ)** judges the consensus states:
   *Full consensus:* $p_s^k > \frac{2}{3}$ and $p_b^k > 0.8$         ▷ *Equation* (2)
   *Partial consensus:* $p_s^k \geq \frac{2}{n}$ and $p_b^k > 0.5$         ▷ *Equation* (3)
   *No consensus:* Other conditions
   **if** Reaching "partial consensus" **then**
      **Collaborator assignment (CA)** selects the collaborators:
      Calculate the conflict score for $p$-th and $q$-th opinion groups $\psi_{pq} = \psi_{pq}^{\mathcal{G}} \cdot \psi_{pq}^{\mathcal{L}}$ ▷ *Equation* (5)
      **for** Each $a_j \in A$ **do**
         **if** $a_j$ is the agent with lowest belief in most uncertain group $G_u$ **then**
            Select the agents with highest beliefs in conflicting opinion groups ($\psi_{pq} > 2$)
         **else**
            Select the agents with highest beliefs in supportive opinion groups ($\psi_{pq} \leq 2$)
         **end if**
      **end for**
   **else if** Reaching "no consensus" **then**
      **Leader selection (LS)** selects the leaders:
      **for** Each opinion group **do**
         Select $n^l$ agents with highest beliefs in the opinion group
      **end for**
   **end if**
**until** Reaching "full consensus" or maximum iteration number

---

# B   Proofs of Theorems

**Theorem 3.2.** *Let $\{x_i^k\}_{i=1}^n$ denote the opinions and $\{b_i^k\}_{i=1}^n$ denote the beliefs of a MAS with $n$ agents at the $k$-th step of collaboration. The collaboration between agents satisfies the following properties:*

*1. When each agent in MAS collaborates with supportive agents, the MAS tends to reach the stable consensus, converging to the state of the average opinion and belief of all collaborating agents.*

2. *When any agent in MAS collaborates with conflicting agents, the MAS tends to form the unstable consensus, potentially leading to divergence or oscillation in group states.*

*Proof.* Firstly, when $i$-th agent collaborates with the supportive agents, its transmission of opinion $x_i^{k+1}$ and the average opinion of the collaborative agents $\bar{x}^k$ at $k$-th step are represented as Equation (7) according to the definitions in Section 2.1.

$$x_i^{k+1} = (1 - n_i^s \alpha) x_i^k + \sum_{a_j \in A_i^s} \alpha x_j^k, \qquad \bar{x}^k = \frac{1}{n_i^s} \sum_{a_j \in A_i^s} x_j^k. \tag{7}$$

The transmission of belief $b_i^{k+1}$ and the average belief of the collaborative agents $\bar{b}^k$ at $k$-th step are represented as Equation (8) according to the definitions in Section 2.1.

$$b_i^{k+1} = (1 - n_i^s \beta) b_i^k + \sum_{a_j \in A_i^s} \beta b_j^k, \qquad \bar{b}^k = \frac{1}{n_i^s} \sum_{a_j \in A_i^s} b_j^k. \tag{8}$$

For each $i \in [1, n]$, the increment in the distance between $i$-th agent's opinion and the average opinion of the collaborative agents before and after the $k$-th collaboration step is represented as Equation (9).

$$
\begin{aligned}
& (x_i^{k+1} - \bar{x}^k)^2 - (x_i^k - \bar{x}^k)^2 \\
= & ((1 - n_i^s \alpha) x_i^k + \sum_{a_j \in A_i^s} \alpha x_j^k - \bar{x}^k)^2 - (x_i^k - \bar{x}^k)^2 \\
= & (x_i^k - \bar{x}^k + \sum_{a_j \in A_i^s} \alpha x_j^k - n_i^s \alpha x_i^k)^2 - (x_i^k - \bar{x}^k)^2 \\
= & ((x_i^k - \bar{x}^k)(1 - \alpha n_i^s))^2 - (x_i^k - \bar{x}^k)^2 = [(1 - \alpha n_i^s)^2 - 1](x_i^k - \bar{x}^k)^2 \le 0.
\end{aligned}
\tag{9}
$$

For each $i \in [1, n]$, the increment in the distance between $i$-th agent's belief and the average belief of the collaborative agents before and after the $k$-th collaboration step is represented as Equation (10).

$$
\begin{aligned}
& (b_i^{k+1} - \bar{b}^k)^2 - (b_i^k - \bar{b}^k)^2 \\
= & ((1 - n_i^s \beta) b_i^k + \sum_{a_j \in A_i^s} \beta b_j^k - \bar{b}^k)^2 - (b_i^k - \bar{b}^k)^2 \\
= & (b_i^k - \bar{b}^k + \sum_{a_j \in A_i^s} \beta b_j^k - n_i^s \beta b_i^k)^2 - (b_i^k - \bar{b}^k)^2 \\
= & ((b_i^k - \bar{b}^k)(1 - \beta n_i^s))^2 - (b_i^k - \bar{b}^k)^2 = [(1 - \beta n_i^s)^2 - 1](b_i^k - \bar{b}^k)^2 \le 0.
\end{aligned}
\tag{10}
$$

Due to $0 \le n_i^s \le n$ and $\alpha = \beta = \frac{2}{n}$, thus the conditions "$\le 0$" in Equation (9) and (10) hold. Therefore, when collaborates with the supportive agents, the opinion and belief of $i$-th agent converge to the average opinion and belief of collaborators gradually, thus when each agent collaborates with supportive agents, the opinions and beliefs of MAS converge, which can reach the stable consensus. When the incremental values in the Equation (9) and (10) become to 0 for each $i \in [0, n]$, the opinions and beliefs of multi-agent system converge to the average opinion and belief of the collaborative agents.

Secondly, when $i$-th agent collaborates with the conflicting agents, its transmission of opinion $x_i^{k+1}$ and the average opinion of the collaborative agents $\bar{x}^k$ are represented in Equation (11) according to the definitions in Section 2.1.

$$x_i^{k+1} = (1 - n_i^c \alpha) x_i^k + \sum_{a_j \in A_i^c} \alpha x_j^k, \qquad \bar{x}^k = \frac{1}{n_i^c} \sum_{a_j \in A_i^c} x_j^k. \tag{11}$$

The transmission of belief $b_i^{k+1}$ and the average belief of the collaborative agents $\bar{b}^k$ are represented in Equation (12) according to the definitions in Section 2.1.

$$b_i^{k+1} = (1 + n_i^c \beta) b_i^k - \sum_{a_j \in A_i^c} \beta b_j^k, \qquad \bar{b}^k = \frac{1}{n_i^c} \sum_{a_j \in A_i^c} b_j^k. \tag{12}$$

For each $i \in [1, n]$, the increment in the distance between $i$-th agent's opinion and the average belief of the collaborative agents before and after the collaboration is represented as Equation (13).

$$
\begin{aligned}
&(x_i^{k+1} - \bar{x}^k)^2 - (x_i^k - \bar{x}^k)^2 \\
&= ((1 - n_i^c \alpha) x_i^k + \sum_{a_j \in A_i^s} \alpha x_j^k - \bar{x}^k)^2 - (x_i^k - \bar{x}^k)^2 \\
&= ((x_i^k - \bar{x}^k)(1 - n_i^c \alpha))^2 - (x_i^k - \bar{x}^k)^2 = [(1 - \alpha n_i^c)^2 - 1](x_i^k - \bar{x}^k)^2 \le 0.
\end{aligned}
\tag{13}
$$

For each $i \in [1, n]$, the increment in the distance between $i$-th agent's belief and the average belief of the collaborative agents before and after the collaboration is represented as Equation (14).

$$
\begin{aligned}
&(b_i^{k+1} - \bar{b}^k)^2 - (b_i^k - \bar{b}^k)^2 \\
&= ((1 + n_i^c \beta) b_i^k - \sum_{a_j \in A_i^s} \beta b_j^k - \bar{b}^k)^2 - (b_i^k - \bar{b}^k)^2 \\
&= ((b_i^k - \bar{b}^k)(1 + n_i^c \beta))^2 - (b_i^k - \bar{b}^k)^2 = [(1 + \beta n_i^c)^2 - 1](b_i^k - \bar{b}^k)^2 \ge 0.
\end{aligned}
\tag{14}
$$

Due to $0 \le n_i^s \le n$, thus the "$\le 0$"in Equation (13) and the "$\ge 0$"in Equation (14) hold. Therefore, when any agent collaborates with the conflicting agents, the opinion of $i$-th agent converge to the average opinion gradually, thus the multi-agent system (MAS) can reach the consensus. However, the belief of $i$-th agent can not converge to the average belief, thus MAS can not reach a stable consensus. $\qquad \square$

**Theorem 3.3.** *Let $\{x_i^k\}_{i=1}^n$ and $\{b_i^k\}_{i=1}^n$ represent the opinions and beliefs of a MAS with $n$ agents at the $k$-th step. Within each opinion group, the $i$-th agent follows $n^l$ leaders, and the collaboration between followers and their respective leaders satisfies the following properties:*

1. *When each agent in an opinion group collaborates with its leaders, the MAS tends to reach the stable consensus, converging to the average state of the leaders.*
2. *When the leaders' average belief is higher than other agents' beliefs, the leaders with higher beliefs can expedite the convergence to the stable consensus.*

*Proof.* Firstly, when the $i$-th agent collaborates with the leaders in one opinion group, since different opinions within the same opinion group share similar topics, they support each other, its transmission of opinion/belief and the average opinion/belief of the collaborative agents at $k$-th step are represented as Equation (15)/(16) which are similar to Equation (7)/(8).

$$
x_i^{k+1} = (1 - n_i^s \alpha) x_i^k + \sum_{a_j \in A^l} \alpha x_j^k, \qquad \bar{x}^k = \frac{1}{n_i^s} \sum_{a_j \in A^l} x_j^k.
\tag{15}
$$

$$
b_i^{k+1} = (1 - n_i^s \beta) b_i^k + \sum_{a_j \in A^l} \beta b_j^k, \qquad \bar{b}^k = \frac{1}{n_i^s} \sum_{a_j \in A^l} b_j^k.
\tag{16}
$$

For each $i \in [1, n]$ If the $i$-th agent is a follower, the increment in the distance between $i$-th agent's opinion and the average opinion of the leaders at $(k+1)$-th and $k$-th steps are shown as Equation (17).

$$
\begin{aligned}
\Delta_x^{k+1} &= \| \frac{1}{n^l} \sum_{a_j \in A^l} x_j^k - x_i^{k+1} \| - \| \frac{1}{n^l} \sum_{a_j \in A^l} x_j^k - x_i^k \| \\
&= \| \frac{1}{n^l} \sum_{a_j \in A^l} x_j^k - ((1 - n^l \alpha) x_i^k + \sum_{a_j \in A^l} \alpha x_j^k) \| - \| \frac{1}{n^l} \sum_{a_j \in A^l} x_j^k - x_i^k \| \\
&= \| \frac{1}{n^l} \sum_{a_j \in A^l} x_j^k - x_i^k + n^l \alpha x_i^k - \sum_{a_j \in A^l} \alpha x_j^k \| - \| \frac{1}{n^l} \sum_{a_j \in A^l} x_j^k - x_i^k \| \\
&= \| (\frac{1}{n^l} - \alpha)(\sum_{a_j \in A^l} x_j^k - n^l x_i^k) \| - \| \frac{1}{n^l} (\sum_{a_j \in A^l} x_j^k - n^l x_i^k) \| \\
&= (| \frac{1}{n^l} - \alpha | - \frac{1}{n^l}) \| \sum_{a_j \in A^l} x_j^k - n^l x_i^k \| \le 0.
\end{aligned}
\tag{17}
$$

The increment in the distance between $i$-th agent's belief and the average belief of the leaders at $(k+1)$-th and $k$-th steps are shown as Equation (18).

$$\Delta_b^{k+1} = \|\frac{1}{n^l}\sum_{a_j \in A^l} b_j^k - b_i^{k+1}\| - \|\frac{1}{n^l}\sum_{a_j \in A^l} b_j^k - b_i^k\|$$

$$= \|\frac{1}{n^l}\sum_{a_j \in A^l} b_j^k - ((1 - n^l\beta)b_i^k + \sum_{a_j \in A^l} \beta b_j^k)\| - \|\frac{1}{n^l}\sum_{a_j \in A^l} b_j^k - b_i^k\|$$

$$= \|\frac{1}{n^l}\sum_{a_j \in A^l} b_j^k - b_i^k + n^l\beta b_i^k - \sum_{a_j \in A^l} \beta b_j^k\| - \|\frac{1}{n^l}\sum_{a_j \in A^l} b_j^k - b_i^k\| \quad (18)$$

$$= \|(\frac{1}{n^l} - \beta)(\sum_{a_j \in A^l} b_j^k - n^l b_i^k)\| - \|\frac{1}{n^l}(\sum_{a_j \in A^l} b_j^k - n^l b_i^k)\|$$

$$= (|\frac{1}{n^l} - \beta| - \frac{1}{n^l})\|\sum_{a_j \in A^l} b_j^k - n^l b_i^k\| \le 0.$$

Similarly, if the $i$-th agent is a leader, $\Delta_x^{k+1} = (|\frac{1}{n^l-1} - \alpha| - \frac{1}{n^l-1})\|\sum_{a_j \in A^l} x_j^k - (n^l - 1)x_i^k\|$, $\Delta_b^{k+1} = (|\frac{1}{n^l-1} - \beta| - \frac{1}{n^l-1})\|\sum_{a_j \in A^l} b_j^k - (n^l - 1)b_i^k\|$. Due to $0 < \alpha \le \frac{2}{n} < \frac{2}{n^l} < \frac{2}{n^l-1}$, the value of $\Delta_x^{k+1}$ and $\Delta_b^{k+1}$ are not larger than 0 consistently, in which $\Delta_x^{k+1} = 0$ and $\Delta_b^{k+1} = 0$ indicate the opinion and belief of leader or follower converge to the average opinion and belief of all leaders.

Secondly, given the condition that the belief $b_i^k$ is globally continuous on all tokens of generated opinions $x_i^k$ [52], we can conclude that if $b_i^k$ does not converge, then $x_i^k$ can not converge either. Therefore the convergence rate of $b_i^k$ determines the consensus rate of multi-agent system (MAS).

The convergence rate is represented by $|\Delta_b^{k+1}|$, which indicates the absolute value of increment in the distance between the $i$-th agent's belief and leaders' average belief from step $k$ to $k + 1$. The comparison of the convergence rate between $|\Delta_b^{k+1}|$ with higher belief $b_j^k$ and $|\Delta_b^{k+1}|'$ with lower belief $(b_j^k)'$ is shown in Equation (19).

$$|\Delta_b^{k+1}| - |\Delta_b^{k+1}|' = |w_i^k|(\|\sum_{a_j \in A^l} b_j^k - n^l b_i^k\| - \|\sum_{a_j \in (A^l)'} (b_j^k)' - n^l b_i^k\|) > 0, \quad (19)$$

where $w_i^k = (|\frac{1}{n^l} - \beta| - \frac{1}{n^l})$ when $i$-th agent is a follower, and $w_i^k = (|\frac{1}{n^l-1} - \beta| - \frac{1}{n^l-1})$ when $i$-th agent is a leader. Due to the average belief of leaders is higher than other agents' beliefs, it can derive that the condition of "$> 0$" holds, thus the convergence rate is higher when collaborating with leaders who consistently maintain higher beliefs. $\square$

## C Experimental Details

### C.1 Baseline Details

- **CoT** [20] is a single-agent reasoning method, which conducts reasoning step-by-step.
- **Reflection** [21] is a single-agent reasoning method, which reflects on their errors and apply self-directed strategies to strengthen the solutions.
- **CoT-SC** [22] is a single-agent reasoning method, which samples multiple reasoning paths and select the majority result.
- **EoT** [23] is a collaboration method, in which each agent can receive opinion from its predecessor and send its own opinion to the next agent.
- **GroupDebate** [24] is a collaboration method, which conducts internal discussions first and then summarizes the results as the input for all agents in the next step.
- **MAD** [18] is a collaboration method, which enhances solutions through multi-agent debate to refine the answer.
- **PARSE** [25] is a collaboration method, which conducts multi-agent collaboration with sparse collaboration structure.

- **CMD** [26] is a collaboration method, in which agents within the same group receive solutions with explanations, while those in different groups receive solutions without explanations.
- **DyLAN** [3] is a collaboration method, which selects agents based on their contributions to problem-solving.

## C.2 Benchmark Datasets Details

- **MATH** [8] is a mathematical reasoning benchmark that contains 5,000 cases covered 7 types of problems, including algebra, counting and probability, geometry, intermediate algebra, number theory, prealgebra and precalcus. MATH dataset is released under *MIT License*, which can be found in https://huggingface.co/datasets/HuggingFaceTB/MATH.
- **MMLU** [9] is an integrated reasoning benchmark that contains 57 subjects covered by the 4 main types of problems, including STEM, social sciences, humanities and other. MMLU dataset is released under *MIT License*, which can be found in https://people.eecs.berkeley.edu/~hendrycks/data.tar.

The results of full data in Figure 4(e) are sourced from [53] and https://crfm.stanford.edu/helm/mmlu/latest/, respectively. All datasets are from public sources, ethically reviewed by publishers, and the cases have undergone anonymization to safeguard sensitive information.

## C.3 Model Details

In the experiments, we use the large language models (LLMs) with different parameter sizes as backbone models of agents, the details of the models are listed as follows:

- **Qwen2.5-7B-Instruct** [53] is released under *Apache license 2.0*, which can be found in https://huggingface.co/Qwen/Qwen2.5-7B-Instruct.
- **Qwen2.5-14B-Instruct** [53] is released under *Apache license 2.0*, which can be found in https://huggingface.co/Qwen/Qwen2.5-14B-Instruct.
- **Qwen2.5-32B-Instruct** [53] is released under *Apache license 2.0*, which can be found in https://huggingface.co/Qwen/Qwen2.5-32B-Instruct.
- **DeepSeek-R1-Distill-Qwen-32B** [54] is released under *MIT License*, which can be found in https://huggingface.co/deepseek-ai/DeepSeek-R1-Distill-Qwen-32B.

## C.4 Implementation Details

**Experimental Setting**   To ensure the opinion diversity within MAS, we set the *temperature* as 0.7. The number of leaders $n^l$ is set as 2 and to ensure that at least one opinion group contains more than $n^l$ agents, and to allow for the potential coexistence of supportive and conflicting relationships among opinion groups, thus we set the number of opinion group clustering for KMeans [55] in BCCS as 3. The TF-IDF vectors capture the keyword distributions to represent the topics [56], enabling KMeans clustering based to topical similarity. When an opinion group's size is at most $n^l$, each agent updates their opinion based on all group members' opinions. The answer probability is calculated by multiplying the token probabilities of the final answer sentence. The prompts are listed in Table 15.

**Computer Resources**   All experiments are conducted with Nvidia A800 GPUs with 80GB memory. Specifically, for experiments based on 7B and 14B models, the experiments need one A800 GPU, and the experiments based on 32B models, the experiments need two A800 GPUs. The average execution time is about 1 minute per MMLU case and 3 minutes per MATH case.

# D  Supplementary Experiments

## D.1  Additional Scenarios of NLP Tasks

We evaluate on two additional NLP benchmark datasets with two reasoning scenarios, including FOLIO [57] of logical reasoning and CommonsenseQA [58] of commonsense reasoning. We compare our proposed BCCS with three best performed baselines in Table 1 and Table 2 and we randomly select 62 cases from FOLIO and 92 cases from CommonsenseQA for comparison, the results are listed in Table 5, which demonstrate that our proposed BCCS performs better than the baselines on the two scenarios.

Table 5: Main results on the FOLIO and CommonsenseQA datasets.

| Dataset | CMD | MAD | PARSE | BCCS |
|---|---|---|---|---|
| FOLIO | 79.03 | 80.65 | 77.42 | **82.26** |
| CommonsenseQA | 79.35 | 78.26 | 80.43 | **82.61** |

## D.2 Analysis of Hyperparameters

In this section, we analyze the performance of the BCCS under different hyperparameters settings. Specifically, we analyze the impact of three hyperparameters, including agent number $n$, maximum rounds, leader number $n^l$, and randomly select 500 cases from MATH and MMLU for analysis.

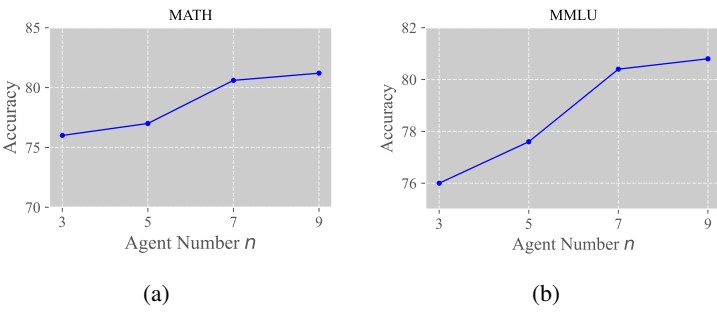

(a)                                          (b)

Figure 5: The performance of BCCS with different agent numbers $n$.

**Effect of Agent Number**    We evaluate the performance of BCCS with agent numbers ranging in $\{3, 5, 7, 9\}$. The results are shown in Figure 5, as the agent number $n$ increases, the accuracy of BCCS improves. The performance gap between $n = 7$ and $n = 9$ is not significant. The results show that using seven agents ($n = 7$) achieves an optimal balance between performance and efficiency in BCCS's execution as the execution efficiency declines with the number of agents increases.

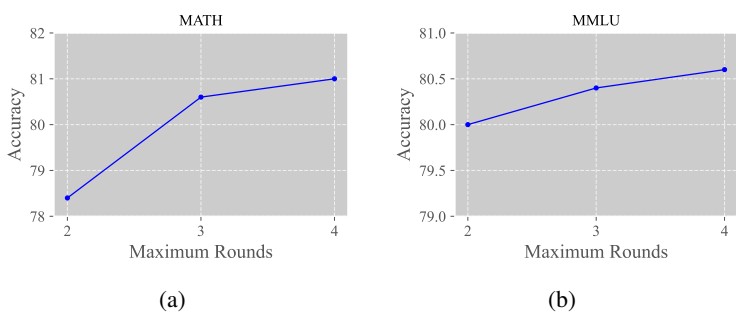

(a)                                          (b)

Figure 6: The performance of BCCS with different maximum rounds.

**Effect of Maximum Rounds**    We evaluate the performance of BCCS with maximum rounds ranging in $\{2, 3, 4\}$. The results are shown in Figure 6, as the number of maximum rounds increases, the accuracy of BCCS improves. The performance of BCCS between 3 and 4 rounds is not significant. The results indicate that a maximum of 3 rounds achieves the optimal balance between performance and efficiency in BCCS's execution, as the execution efficiency declines with the rounds increase.

**Effect of Leader Number**    We evaluate the performance of BCCS with the number of leaders $n^l$ ranging in $\{1, 2, 3\}$. The results are shown in Figure 7, as the number of leaders increases, the accuracy of BCCS increases first and then descends, which increments are not significant. The reason may be that two leaders can balance individual errors and excessive divergence, ensuring the system converges correctly.

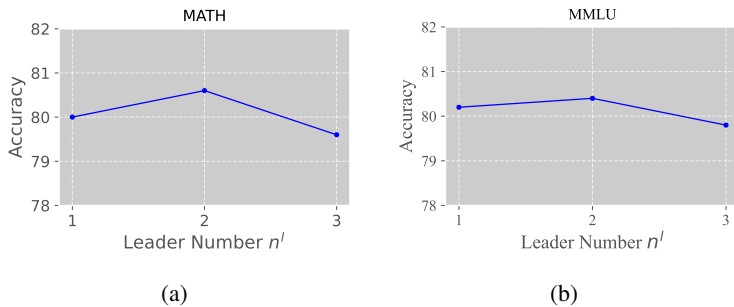

Figure 7: The performance of BCCS with different leader numbers $n^l$.

Besides, we also analyze the impact of three hyperparameters which optimal values are derived mathematically in Section 4, including the thresholds for full consensus, partial consensus and conflicting score, and randomly select 140 cases from MATH and 114 cases from MMLU for analysis.

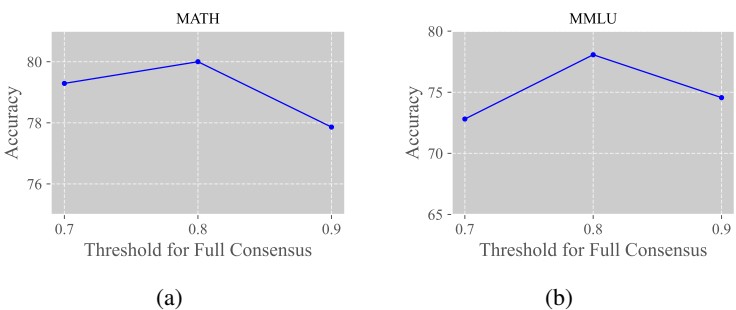

Figure 8: The performance of BCCS with different thresholds for full consensus.

**Effect of Threshold for Full Consensus**    To demonstrate the sensitivity of the threshold for full consensus, we compare the performance of different values for the thresholds of full consensus ranging in $\{0.7, 0.8, 0.9\}$, as shown in Figure 8. The results show that the mathematically derived values of the threshold for full consensus as shown in Section 4.1 are optimal consistently.

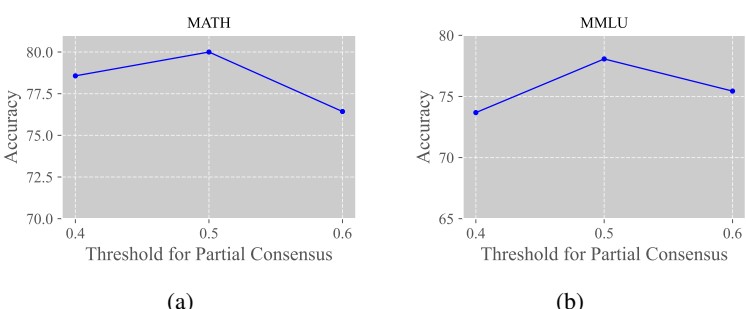

Figure 9: The performance of BCCS with different thresholds for partial consensus.

**Effect of Threshold for Partial Consensus**    To demonstrate the sensitivity of the threshold for partial consensus, we compare the performance of different values for the thresholds of partial consensus ranging in $\{0.4, 0.5, 0.6\}$, as shown in Figure 9. The results show that the mathematically derived values of the threshold for partial consensus as shown in Section 4.1 are optimal consistently.

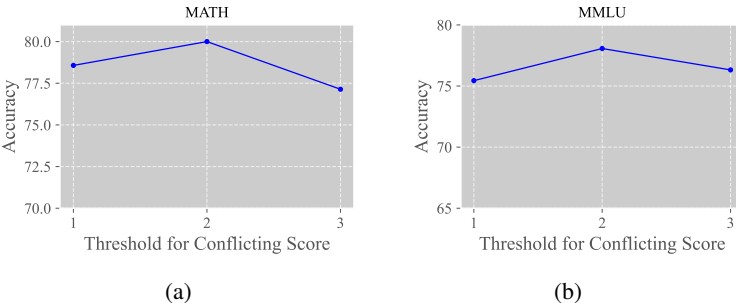

Figure 10: The performance of BCCS with different thresholds for conflicting score.

**Effect of Threshold for Conflicting Score** $\psi_{pq}$    To demonstrate the sensitivity of the threshold for conflicting score $\psi_{pq}$, we compare the performance of different values for the thresholds of conflicting score $\psi_{pq}$ ranging in $\{1, 2, 3\}$, as shown in Figure 10. The results show that the mathematically derived values of the threshold for conflicting score $\psi_{pq}$ as shown in Section 4.2 are optimal consistently.

### D.3    Analysis of Different Model Sizes

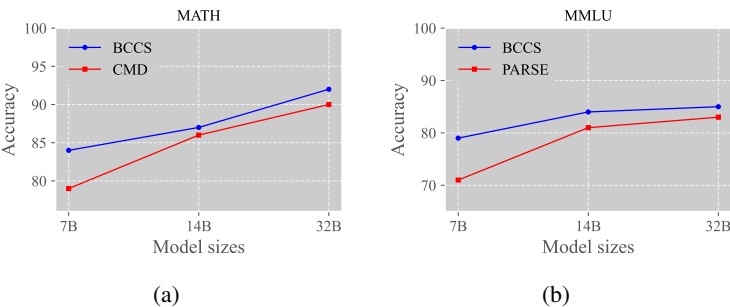

Figure 11: The comparison between BCCS and the strongest baselines on MATH and MMLU datasets with model sizes of 7B, 14B and 32B.

In this section, we compare the performance of the BCCS with the state-of-the-art baselines based on different model sizes, including 7B, 14B and 32B on MATH and MMLU, using 100 randomly sampled cases for each dataset. Specifically, we use Qwen2.5-7B-Instruct, Qwen2.5-14B-Instruct and Qwen2.5-32B-Instruct as backbone models of BCCS and CMD on MATH, Qwen2.5-7B-Instruct, Qwen2.5-14B-Instruct and DeepSeek-R1-Distill-Qwen-32B as backbone models of BCCS and PARSE on MMLU.

For all sizes of models, BCCS outperforms the state-of-the-are baselines consistently, which demonstrate the generalization capability of BCCS across models of varying sizes. For the smaller backbone model with 7B parameters, BCCS delivers significant improvements, demonstrating its ability to maintain strong performance with fewer computational requirements.

### D.4    Analysis of Computational Scalability

To better understand the computational scalability, we randomly select 500 cases from MATH and MMLU for analysis and separately count the average number of tokens per case (denoted by "#Token") as an estimate of computational cost for both our proposed BCCS and the SOTA baseline on each dataset of MATH and MMLU, as shown in Table 6. The results demonstrate that BCCS achieves better performance with lower computational cost. Besides, although the conflict score calculation requires additional time, it is a lightweight operation that does not require GPU resources, thus the computation time is practically negligible.

Table 6: Anslysis results of computational scalability on the MATH and MMLU datasets.

| MATH | BCCS | CMD |
|------|------|-----|
| #Token | 6554 | 9224 |

| MMLU | BCCS | PARSE |
|------|------|-------|
| #Token | 2981 | 6349 |

## D.5 Analysis of Performance in Adversarial Scenarios

Table 7: The performance in the adversarial scenarios on the MATH and MMLU datasets.

| MATH | BCCS | AdvNoise | CMD |
|------|------|----------|-----|
| Accuracy | 80.00 | 79.29 | 78.57 |

| MMLU | BCCS | AdvNoise | PARSE |
|------|------|----------|-------|
| Accuracy | 78.07 | 76.32 | 70.18 |

To simulate adversarial and noisy scenarios, we conduct an analysis experiment by misreporting beliefs. Specifically, we introduce adversarial conditions of misreporting beliefs by perturbing the belief of one randomly selected agent in each round (denoted as "AdvNoise"), either by increasing lower beliefs or decreasing higher beliefs. In experiments, we randomly select 140 cases from MATH and 114 cases from MMLU for analysis. The results are shown in Table 7, which demonstrate that misreporting beliefs can lead to some performance degradation, yet the overall performance remains higher than the SOTA baseline on each dataset. This is because the remaining correct beliefs are still capable of calibrating the inaccurate or misreported answers, demonstrating the robustness of our proposed BCCS in adversarial scenarios involving noisy agents.

## D.6 Analysis of Performance with Heterogeneous Backbones

Table 8: The performance with heterogeneous backbones on the MATH and MMLU datasets.

| MATH | BCCS | CMD |
|------|------|-----|
| Accuracy | **77.86** | 73.57 |

| MMLU | BCCS | PARSE |
|------|------|-------|
| Accuracy | **72.81** | 68.42 |

To demonstrate the effectiveness of our approach with heterogeneous backbones, we utilize Qwen2.5-7B-Instruct, Phi-3-mini-4k-Instruct (3.8B), and Llama-3.2-1B-Instruct as backbones. In experiments, we randomly select 140 cases from MATH and 114 cases from MMLU for analysis. The results are presented in Table 8, which showcase that our proposed BCCS outperforms the SOTA baseline on each dataset when applied to heterogeneous backbones.

## D.7 Case Study

**Effect of Collaborator Assignment (CA)** Table 9 shows a case study demonstrating the effectiveness of collaborator assignment (CA) module in the BCCS. In this case, BCCS convergences to consensus with correct answer, while the strongest baseline PARSE converges to the suboptimal consensus with incorrect answer. For both BCCS and PARSE in Round 1, Agent 1 and Agent 2 generate a correct and an incorrect answer respectively.

For BCCS in the Round 2, Agent 1 collaborates with its conflicting collaborator, Agent 2, and updates its answer to the correct one. Agent 2 receives its own correct opinion, which is self-supporting, and maintains the correct answer.

For PARSE in Round 2, both Agent 1 and Agent 2 receive all opinions from Round 1. Agent 1 adopts the stubborn strategy [14] to remain its own incorrect answer and Agent 2 adopts the suggestive strategy [14] to follow the Agent 1 to update its answer to the incorrect one, thus PARSE converges to suboptimal consensus with incorrect answer.

**Effect of Leader Selection (LS)**    Table 10 shows a case study demonstrating the effectiveness of leader selection (LS) module in the BCCS. In this case, BCCS convergences to consensus with correct answer, while the baseline MAS converges to the suboptimal consensus with incorrect answer. For both BCCS and MAD in Round 1, Agent 1 and Agent 2 generate a correct and an incorrect answer respectively.

For BCCS in the Round 2, Agent 1 follow the leader Agent 2 to update its answer to the correct one. Agent 2 is selected as the leader, and it remains its own correct answer.

For MAD in the Round 2, both Agent 1 and Agent 2 receive all opinions from round 1. Agent 1 adopts the stubborn strategy [14] to remain its own incorrect answer and Agent 2 adopts the suggestive strategy [14] to follow the Agent 1 to update its answer to the incorrect one, thus MAD converges to suboptimal consensus with incorrect answer.

**Effect of Belief-Calibrated Consensus Judgment (BCCJ)**    Table 11 shows a case study demonstrating the effectiveness of belief-calibrated consensus judgment (BCCJ) module in the BCCS. In this case, "-BCCJ" indicates replacing the BCCJ with *Byzantine consensus* judgment method. BCCS convergences to consensus with correct answer, while the baseline "-BCCJ" outputs the incorrect answer. For both BCCS and "-BCCJ" in Round 1, the major results (the most frequent voting results in Table 11) are incorrect. "-BCCJ" judges that $p_s^1 = 0.86 > \frac{2}{3}$, thus terminating the collaboration and outputs the incorrect answer.

BCCS in Round 1 judges that $p_s^1 = 0.57 < \frac{2}{3}$ and $p_b^1 = 0.13 < 0.5$, thus it reaches the state of *no consensus*. In the Round 2, BCCS selects the leaders for collaboration and updates the major result to the correct one. Both $p_s^2$ and $p_b^2$ improve from Round 1, reaching $p_s^2 = 0.86 > \frac{2}{3}$ and $p_b^2 = 0.97 > 0.8$ from Round 1, which indicates the system reaches *full consensus*, yielding correct answer.

**Error Analysis**    We demonstrate the error cases of BCCS in Table 12 and Table 13. In Table 12, Leader 1 and Follower generate a correct answer and Leader 2 generates an incorrect answer in Round 1. In Round 2, Follower receives the opinions from Leader 1 and Leader 2 in Round 1, the disagreement between the two leaders leads the follower to modify its own original opinion to align with the two leaders' opinions, thus ultimately shifting from the correct answer in Round 1 to the incorrect answer in Round 2. The case in Table 13 demonstrates that BCCS is able to correctly identify the incorrect result produced by the agent as unreliable and appropriately selects a conflicting agent as the collaborator. However, due to limitations in model performance, all agents fail to generate the correct answer, thus the agent can not update a correct answer in the next round.

Table 9: Case study of collaborator assignment (CA), (✓) indicates correct answer and (✗) indicates incorrect answer.

**Question:** The encomienda system was used during the colonization of the Americas to regulate the indigenous people, was not ended by which of the following?: A) The protests of the Catholic missionaries against abuses of forced labor, B) The lack of new land to assign to well-connected Spaniards and conquistadores, C) The Spanish royal crown's desire to control the estates more directly, D) An increase in the number of mestizos, who by law were forbidden from working on the encomiendas.

| Method | Round | Opinions | Results |
|---|---|---|---|
| BCCS | 1 | **Agent 1**: No evidence of an increase in the number of mestizos or any specific laws forbidding them from working on encomiendas. The answer is D. | D (✗) |
| | | **Agent 2**: There was still land available for assignment, as the encomienda system required land grants to manage indigenous populations. The answer is B. | B (✓) |
| | 2 | **Agent 1**: (Receive conflicting opinion from **Agent 2** in round 1) There was still land available for assignment, as the encomienda system required land grants to manage indigenous populations. The answer is B. | B (✓) |
| | | **Agent 2**: (Receive own supporting opinion from **Agent 2** in round 1) There was still land available for assignment, as the encomienda system required land grants to manage indigenous populations. The answer is B. | B (✓) |
| PARSE | 1 | **Agent 1**: It is more about the demographic changes and social dynamics rather than a direct cause for the termination of the encomienda system. The answer is D. | D (✗) |
| | | **Agent 2**: There was still land available to assign, at least in the short term. Therefore, the lack of new land would not have been a reason to end the system. The answer is B. | B (✓) |
| | 2 | **Agent 1**: (Receive opinions from **Agent 1** and **Agent 2** in round 1) The most accurate answer is that the increase in the number of mestizos, who by law were forbidden from working on the encomiendas. The answer is D. | D (✗) |
| | | **Agent 2**: (Receive opinions from **Agent 1** and **Agent 2** in round 1) The least likely reason is the increase in the number of mestizos, who by law were forbidden from working on the encomiendas. The answer is D. | D (✗) |

Table 10: Case study of leader selection (LS), (✓) indicates correct answer and (✗) indicates incorrect answer.

**Question:** A scientist, using electrodes, is stimulating a group of neurons in the hypothalamus and recording their membrane potential changes. She observes a sharp rise in membrane potential when she first stimulates them, the the difference of 100mV. When she tries another stimulation immediately after the first, there is no response. However, when she increase the amount of voltage she supplies to the second shock, she observes a similar response to the first. What time period of stimulation would the second shock be characterized in regards to membrane potential?: A) Depolarization, B) Repolarization, C) Hyperpolarization, D) Resting potential.

| Method | Round | Opinions | Results |
|--------|-------|----------|---------|
| BCCS | 1 | **Agent 1**: Given that the neuron responded to a stronger stimulus but not to a normal one, it suggests that the membrane potential was still moving towards the resting state but had not fully reached it. The answer is B. | B (✗) |
| | | **Agent 2**: Given that the neuron showed a response only with increased voltage, indicating it was not at the resting potential, but rather in a state where it required more stimulus to reach the threshold for an action potential, the most appropriate term is hyperpolarization. The answer is C. | C (✓) |
| | 2 | **Agent 1**: (Receive the opinion from the leader **Agent 2** in the round 1) Given the context and the need for the neuron to require more stimulus to respond, the most accurate characterization of the second shock is hyperpolarization. The answer is C. | C (✓) |
| | | **Agent 2**: (**Agent 2** is selected as the leader) Given the observations and the need for increased voltage to achieve a response, the correct characterization of the membrane potential during the second shock is hyperpolarization. The answer is C. | C (✓) |
| MAD | 1 | **Agent 1**: The most likely explanation is that the neurons underwent repolarization and then hyperpolarization after the initial depolarization. The answer is B. | B (✗) |
| | | **Agent 2**: The second shock would be characterized by the neuron being in a state of hyperpolarization, which is a state where the membrane potential is more negative than the resting potential. The answer is C. | C (✓) |
| | 2 | **Agent 1**: (Receive opinions from **Agent 1** and **Agent 2** in round 1) It correctly identifies that the second shock occurs during the refractory period, specifically during the phase of repolarization but still within the refractory period where the cell is less responsive. The answer is B. | B (✗) |
| | | **Agent 2**: (Receive opinions from **Agent 1** and **Agent 2** in round 1) Increasing the voltage for the second stimulation allows a similar response, suggesting the neuron has repolarized and is closer to its resting potential but still in a refractory state. The answer is B. | B (✗) |

Table 11: Case study of belief-calibrated consensus judgment (BCCJ), (✓) indicates correct answer and (✗) indicates incorrect answer, and ↑ indicates the corresponding value increases compared to the previous round.

**Question:** Question: What is the magnetic moment of $^{205}$Tl? A) $\gamma(^{205}\text{Tl}) = 1.569 \times 10^8$ T$^{-1}$s$^{-1}$, $I = \frac{1}{2}$, B) $0.717 \times 10^{-26}$ J T$^{-1}$, C) $0.827 \times 10^{-26}$ J T$^{-1}$, D) $1.433 \times 10^{-26}$ J T$^{-1}$.

| Method | Round | Opinions | Major Results |
|---|---|---|---|
| BCCS | 1 | **Agent 1**: … Our calculated value is $0.7845 \times 10^8$ J T$^{-1}$, which is closest to option B when considering the magnitude. The answer is B. **Agent 2**: … The calculated magnetic moment is $0.827 \times 10^{-26}$ J T$^{-1}$, which matches option C. The answer is C. | B (✗) $p_s^1 = 0.57$ $p_b^1 = 0.13$ |
| | 2 | **Agent 1**: (Receive the opinion from the leader **Agent 2** in the round 1) … The calculated magnetic moment ($0.827 \times 10^{-26}$ J T$^{-1}$) matches Option C. The answer is C. **Agent 2**: (**Agent 2** is selected as the leader) … Option C: ($0.827 \times 10^{-26}$ J T$^{-1}$) matches our calculated value. The answer is C. | C (✓) $p_s^2 = 0.86$ ↑ $p_b^2 = 0.97$ ↑ |
| -BCCJ | 1 | **Agent 1**: … The correct value for the magnetic moment should be around ($0.7845 \times 10^8$ J T$^{-1}$), which is closest to option B. The answer is B. **Agent 2**: … The closest option to ($5.197 \times 10^{-26}$ J T$^{-1}$) is ($0.717 \times 10^{-26}$ J T$^{-1}$). The answer is B. | B (✗) $p_s^1 = 0.86$ |

Table 12: Error case analysis of belief-calibrated consensus seeking (BCCS).

**Question:** Which of the following is a true statement about optical isomerism of complexes containing achiral ligands?: A) Square planar complexes can display optical isomerism only if all four ligands are identical, B) Tetrahedral complexes never display optical isomerism, C) Linear complexes can display optical isomerism when both ligands are different, D) Octahedral complexes of monodentate ligands can display optical isomerism only when they have at least three different ligands.

| Round | Opinions | Results |
|---|---|---|
| 1 | **Leader 1**: Octahedral complexes can have optical isomerism through "enantiomers" when they contain chiral centers. For an octahedral complex with monodentate ligands, having at least three different ligands can create chiral centers leading to optical isomerism. However, it is not a strict requirement that there must be at least three different ligands for optical isomerism to occur. The answer is D. | D (✓) |
| | **Leader 2**: Tetrahedral complexes can display optical isomerism if they contain chiral centers . Answer is B. | B (✗) |
| | **Follower**: (D) correctly states that three different ligands can lead to optical isomerism. The answer is D. | D (✓) |
| 2 | **Follower**: (Receive opinions from **Leader 1** and **Leader 2** in round 1) Let's re-examine each option … The (B) provided solution correctly notes that tetrahedral complexes can display optical isomerism if they contain chiral centers … However, geometric isomerism (cis-trans isomerism) can occur with just two different ligands . Therefore, (D) is not entirely accurate. The answer is B. | B (✗) |

Table 13: Error case caused by model performance limitations.

| **Question:** A cannon is mounted on a truck that moves forward at a speed of 5 m/s. The operator wants to launch a ball from a cannon so the ball goes as far as possible before hitting the level surface. The muzzle velocity of the cannon is 50 m/s. At what angle from the horizontal should the operator point the cannon? A) 5°, B) 41°, C) 45°, D) 49°. | |
|---|---|
| Opinions | Results |
| **[Initial Opinion] Agent:** ...the angle that maximizes the range is 45°. | C (✗) |
| **[Initial Opinion] Conflicting Agent:** The optimal angle from the horizontal for the cannon to achieve the maximum range, considering the truck's speed, is approximately 41 degrees. | B (✗) |
| **[Update Opinion] Agent:** The provided solution acknowledges that the truck's speed adds more to the horizontal component at lower angles, leading to an optimal angle of 41 degrees. | B (✗) |

# E  Notations

Table 14: Summary of the main notations.

| Notation | Description |
|---|---|
| $n_{case}$ | The number of cases. |
| $n$ | The total number of agents. |
| $i, j$ | The subscripts for agent, opinion and belief. |
| $A$ | $A$ indicates the agent set, which contains $n$ agents. |
| $a_j \in A$ | $a_j$ indicates $j$-th agent in $A$. |
| $x_j^k, b_j^k$ | The opinion and belief of agent $a_j$ in $k$-th step. |
| $m$ | The total number of opinion groups. |
| $p, q$ | The subscripts for opinion group. |
| $G$ | The opinion group set, which contains $m$ opinions groups. |
| $G_p \in G$ | The $k$-th opinion group in $G$. |
| $G_u$ | The most uncertain opinion group in $G = \{G_p\}_{p=1}^m$. |
| $n^p$ | The number of agents in $G_p$. |
| $a_j \in G_p$ | Agent $a_j$ belongs to the opinion group $G_p$. |
| $u_i^k, v_i^k$ | The outcome increment of opinion and belief for $i$-th agent in $k$-th step. |
| $\alpha, \beta$ | Step sizes for analyzing opinion and belief updating. |
| $A^s, A^c$ | $A^s$ indicates the dominant consensus group, in which contain the largest number of same opinions in $A$. $A^c$ indicates the conflict group, in which the opinions are different from $A^s$. |
| $A_i^s, A_i^c$ | $A_i^s$ is a set of agents which contains the supportive opinions for agent $a_i^k$. $A_i^c$ is a set of agents which contains the conflicting opinions for agent $a_i^k$. |
| $\psi_{pq}$ | The conflict score between opinion group $G_p$ and $G_q$, which considers two aspects of macro- and micro-conflict scores. |
| $\psi_{pq}^{\mathcal{G}}, \psi_{pq}^{\mathcal{L}}$ | $\psi_{pq}^{\mathcal{G}}$ indicates the macro-conflict score and $\psi_{pq}^{\mathcal{L}}$ indicates the micro-conflict score. |
| $p_s^k, p_b^k$ | $p_s^k$ indicates the proportion of the dominant consensus group in $A$. $p_b^k$ indicates the proportion of the beliefs of dominant consensus group in all beliefs of $A$. |
| $u$ | The subscripts of case index. |
| $n_u^s, n_u^r$ | $n_u^s$ indicates the number of consensus agents in $u$-th case. $n_u^r$ indicates the iteration rounds in $u$-th case. |
| $x_u, x_u^*$ | $x_u$ and $x_u^*$ indicate the consensus results and the ground-truth of $u$-th case. |
| $\Theta_p, \Theta_q$ | Local consistency scores of opinion groups $G_p$ and $G_q$. |

# F  Prompts

Table 15: The prompts of BCCS

| MATH | Prompts |
|---|---|
| System | Please reason step by step, and put your final answer within \boxed{}. |
| CA | These are the solutions to the problem from other agents: One supporting agent solution:{}, One conflicting agent solution:{} Selecting and using the trustable solutions from current collaboration as additional information, can you provide your answer to the problem? {Question} |
| LS | These are the solutions to the problem from other agents: One leader solution:{} Selecting and using the leading solutions from current collaboration as additional information, can you provide your answer to the problem? {Question} |

| MMLU | Prompts |
|---|---|
| System | Please reason step by step, and answer the question. |
| CA | Here is the question: {Question} These are the solutions to the problem from other agents: One supporting agent solution {}, One conflicting agent solution {} Judging which solutions are trustable and using the solutions from other agents as additional advice, can you give an updated answer? Examine your solution and that other agents step by step. Put your answer in the form (answer) at the end of your response. (answer) represents choice (A), (B), (C), or (D). |
| LS | Here is the question: {Question} These are the solutions to the problem from other agents: One leader solution {} Judging which solutions can lead the trend of thought and using the solutions from other agents as additional advice, can you give an updated answer? Examine your solution and that other agents step by step. Put your answer in the form (answer) at the end of your response. (answer) represents choice (A), (B), (C), or (D). |

