# OpenReview forum: "Belief-Calibrated Multi-Agent Consensus Seeking for Complex NLP Tasks"
_NeurIPS.cc/2025/Conference — NeurIPS 2025 poster_

### Official Review · Reviewer_npWK · 2025-06-30

**Clarity:** 3
**Significance:** 3
**Originality:** 3
**Rating:** 4
**Confidence:** 3

**Summary:**

This paper investigates how to improve stable consensus in multi-agent systems (MAS) for complex NLP tasks. The authors propose a Belief-Calibrated Consensus Seeking (BCCS) framework which enhances existing consensus methods by explicitly considering agents’ internal beliefs and assigning optimal collaborators or leaders based on belief and conflict analysis. They provide theoretical guarantees under opinion dynamics model, and validate the method on MATH and MMLU datasets, achieving ~2-4% improvements over strong multi-agent baselines.

**Questions:**

How sensitive is BCCS to the hyperparameters like number of leaders or the thresholds in partial consensus detection?

**Ethical Concerns:**

["NO or VERY MINOR ethics concerns only"]

**Final Justification:**

Based on the rebuttal, I recommend that this paper be accepted.

**Quality:**

3

**Strengths And Weaknesses:**

**Strengths**

1. The paper tackles very important problem in multi-agent collaboration with LLMs. It is especially important when addressing unstable or superficial consensus caused by ignoring belief inconsistency.

2. The idea to integrate belief calibration into consensus judgment is novel and nicely formulated.

3. Ablation study is detailed and clearly shows the contribution of each module (BCCJ, CA, LS).

**Weakness**

1. The method needs to compute conflict scores and run iterative consensus update. I am not sure about the computational overhead. The paper does not give detailed time analysis.

2. The proposed belief calibration seems heavily rely on LLM’s output probabilities as belief, but these may not always reflect true uncertainty especially in large-scale LLMs.

3. Experiments are only on MATH and MMLU. More scenarios should be included.

---

> ### Author Rebuttal · Authors · 2025-07-30
>
> We sincerely appreciate your insightful feedback. Our responses below address each point and further clarify our work's contributions. If you have any other questions, please feel free to let us know. Otherwise, if our responses have addressed your comments satisfactorily, we kindly invite you to consider updating the rating.
>
> **Q1.** Analysis of computational overhead.
>
> **A1.** We would like to clarify that although the calculation of conflict score requires additional computation time, it is a lightweight operation, which does not require GPU resources, thus its computation time is practically negligible.
> To better understand the computational efficiency, we separately count the average number of tokens per case (denoted by "#Token") as an estimate of computational cost for both our proposed BCCS and the SOTA baseline on each dataset of MATH and MMLU, as shown in the tables below. The results demonstrate that BCCS achieves better results with lower computational cost.
>
> | MATH | BCCS | CMD |
> |:-----:|:-----:|:-----:|
> | #Token | 6554 | 9224 |
>
> | MMLU | BCCS | PARSE |
> |:-----:|:-----:|:-----:|
> | #Token | 2981 | 6349 |
>
> **Q2.** Probability-based belief representation.
>
> **A2.** Thank you for pointing out the probability may not always reflect true uncertainty. Following prior work [1], we adopt the output probability of LLMs as a proxy for belief. While we acknowledge that LLM's output probabilities may not always perfectly reflect uncertainty, this approximation is a widely used and practical method for belief estimation in large language models.
>
> **Q3.** Performance on more scenarios.
>
> **A3.** Thank you for your constructive suggestion. We have extended the evaluation on two NLP benchmark datasets with two reasoning scenarios, including FOLIO [2] of logical reasoning and CommonsenseQA [3] of commonsense reasoning. We compare our proposed BCCS with three best performed baselines, the results are listed in the table below, which demonstrate that our proposed **BCCS performs better than the baselines on the two datasets**.
>
> |     | FOLIO | CommonsenseQA |
> |:-----:|:-----:|:-----:|
> | CMD | 79.03 | 79.35 |
> | MAD | 80.65 | 78.26 |
> | PARSE | 77.42 | 80.43 |
> | BCCS | **82.26** | **82.61** |
>
> **Q4.** Sensitivity of hyperparameters.
>
> **A4.** Thank you for your insightful comment.
> We would like to clarify that the values of hyperparameters including the consensus and conflict thresholds are derived mathematically, with detailed derivations presented in Section 4. To demonstrate the sensitivity of these thresholds, we compare the performance of different values for the thresholds of full consensus, partial consensus, and conflict scores, as shown in the tables below. The results show that **the mathematically derived values are optimal consistently**.
>
> | Full Consensus   | 0.7   | 0.8   | 0.9  |
> |:-------:|:-------:|:-------:|:-------:|
> | MATH | 79.29 | 80.00 | 77.86 |
> | MMLU | 72.81 | 78.07 | 74.56 |
>
> | Partial Consensus   | 0.4   | 0.5   | 0.6   |
> |:-------:|:-------:|:-------:|:-------:|
> | MATH | 78.57 | 80.00 | 76.43 |
> | MMLU | 73.68 | 78.07 | 75.44 |
>
> | Conflict Score   | 1   | 2   | 3   |
> |:-------:|:-------:|:-------:|:-------:|
> | MATH | 78.57 | 80.00 | 77.14 |
> | MMLU | 75.44 | 78.07 | 76.32 |
>
> For other hyperparameters, including agent number, leader number and rounds, we follow the existing multi-agent methods [4] to select these hyperparameters and provide the comparision of different values of these thresholds in Appendix D.1 to demonstrate the sensitivity of these hyperparameters.
>
> **References**
>
> [1] Chiwei Zhu, Benfeng Xu, Quan Wang, Yongdong Zhang, and Zhendong Mao. On the Calibration of Large Language Models and Alignment. Findings of EMNLP 2023.
>
> [2] Simeng Han, Hailey Schoelkopf, Yilun Zhao, Zhenting Qi, Martin Riddell, Wenfei Zhou and etc. FOLIO: Natural Language Reasoning with First-Order Logic. EMNLP 2024.
>
> [3] Alon Talmor, Jonathan Herzig, Nicholas Lourie, and Jonathan Berant. CommonsenseQA: A Question Answering Challenge Targeting Commonsense Knowledge. NAACL 2019.
>
> [4] Yilun Du, Shuang Li, Antonio Torralba, Joshua B. Tenenbaum, Igor Mordatch. Improving factuality and reasoning in language models through multiagent debate. ICML 2024.

---

> ### Author Response · Authors · 2025-08-04
>
> Thank you for your valuable comments, and we have carefully addressed all of your comments, if there are still any questions requiring further clarification, we sincerely hope you could point them out. If you consider that our revisions have met your expectations, we would be grateful if you could consider raising the evaluation rating accordingly. Your feedback is of utmost importance to us!

---

### Official Review · Reviewer_yocy · 2025-07-01

**Clarity:** 3
**Significance:** 3
**Originality:** 3
**Rating:** 4
**Confidence:** 4

**Summary:**

This paper investigates how to improve consensus-seeking in multi-agent systems for complex natural language processing tasks. It addresses the issues of overlooking agents’ internal beliefs and using non-optimal collaboration strategies. The authors propose a method called Belief-Calibrated Consensus Seeking (BCCS). This method calibrates consensus judgment using agent beliefs and assigns optimal collaborators or leaders based on the state of agreement. The framework is tested on the MATH and MMLU benchmark datasets. The findings show that BCCS outperforms the strongest baseline methods. It improves accuracy by 2.23% on MATH and 3.95% on MMLU.

**Questions:**

See weakness.

**Ethical Concerns:**

["NO or VERY MINOR ethics concerns only"]

**Final Justification:**

Some of my concerns have been addressed. But I also shared similar concerns with other reviewers about the additional computational cost and algorithm sensitivity. So I think this is an OK paper and I will maintain my original score.

**Limitations:**

yes

**Quality:**

3

**Strengths And Weaknesses:**

# Strengths

1. The proposed BCCS framework is well-structured and novel. It integrates three distinct yet complementary modules (BCCJ, CA, LS) that systematically handle different states of consensus, from judgment to dynamic collaborator assignment and leader election.
2. This paper provides the community with valuable insights into building more reliable collaborative multi-agent systems by highlighting the benefits of selective, belief-aware interaction.
3. The paper provides a solid theoretical foundation with formal proofs. This adds rigor and explains why the method works.
4. Extensive experiments on MATH and MMLU, including detailed ablation studies, validate the effectiveness of each module. The framework also demonstrates performance robustness across various model sizes and agent configurations.

# Weaknesses

1. The study's experiments are limited to homogeneous environments where all agents use the identical LLM backbone.  This fails to capture the complex dynamics of heterogeneous groups, where agents with diverse reasoning capabilities and knowledge bases interact. The framework's effectiveness in more realistic, mixed-model scenarios remains unproven. This point should be further discussed.
2. The proposed method introduces significant computational overhead. The paper acknowledges the extra time required but the scalability analysis is insufficient. The authors should provide a dedicated experiment measuring computational costs.
3. The literature review should be enriched by recent work on multi-agent decision-making: Do as We Do, Not as You Think: the Conformity of Large Language Models, ICLR.
4. The definitions for consensus states (Full, Partial, No Consensus) and conflict scores depend on specific, hard-coded thresholds (e.g., $p_b^k > 0.8$). The paper provides insufficient justification for these particular values, and it is unclear how sensitive the system's performance is to them.
5. I would like to see more failure cases and in-depth analysis.
6. Typos: L250: randomlly → randomly; L528: con not → can not

---

> ### Author Rebuttal · Authors · 2025-07-30
>
> We sincerely appreciate your positive feedback on our proposed insights for building reliable collaborative systems, as well as your constructive suggestions. We have carefully addressed each comment below to further clarify our contributions. If you have any other questions, please feel free to let us know. Otherwise, if our responses have addressed your comments satisfactorily, we kindly invite you to consider updating the rating.
>
> **Q1.** Performance with heterogeneous backbones.
>
> **A1.** Thank you for raising an interesting point. Our main experiments follow most current multi-agent works to use identical models as the backbones. To demonstrate the effectiveness of our approach with heterogeneous backbones, we utilize Qwen2.5-7B-Instruct, Phi-3-mini-4k-Instruct (3.8B), and Llama-3.2-1B-Instruct as backbones. The results are presented in the tables below, which showcase that our proposed **BCCS outperforms the SOTA baseline on each dataset when applied to heterogeneous backbones**.
>
> | MATH | BCCS | CMD |
> |:-----:|:-----:|:-----:|
> | Accuracy | **77.86** | 73.57 |
>
> | MMLU | BCCS | PARSE |
> |:-----:|:-----:|:-----:|
> | Accuracy | **72.81** | 68.42 |
>
> **Q2.** Analysis of computational overhead.
>
> **A2.** We would like to clarify that although the calculation of conflict score requires additional computation time, it is a lightweight operation, which does not require GPU resources, thus its computation time is practically negligible.
> To better understand the computational efficiency, we separately count the average number of tokens per case (denoted by "#Token") as an estimate of computational cost for both our proposed BCCS and the SOTA baseline on each dataset of MATH and MMLU, as shown in the tables below. The results demonstrate that BCCS achieves better results with lower computational cost.
>
> | MATH | BCCS | CMD |
> |:-----:|:-----:|:-----:|
> | #Token | 6554 | 9224 |
>
> | MMLU | BCCS | PARSE |
> |:-----:|:-----:|:-----:|
> | #Token | 2981 | 6349 |
>
> **Q3.** Literature review enrichment.
>
> **A3.** We sincerely appreciate your valuable suggestion to enhance our literature review by including the work "Do as We Do, Not as You Think: The Conformity of Large Language Models" (ICLR). We will incorporate this reference in the related work section of the final version.
>
> **Q4.** Justification and sensitivity of consensus and conflict thresholds.
>
> **A4.** Thank you for your insightful comment.
> We would like to clarify that the values of hyperparameters including the consensus and conflict thresholds are derived mathematically, with detailed derivations presented in Section 4. To demonstrate the sensitivity of these thresholds, we compare the performance of different values for the thresholds of full consensus, partial consensus, and conflict scores, as shown in the tables below. The results show that **the mathematically derived values are optimal consistently**.
>
> | Full Consensus   | 0.7   | 0.8   | 0.9  |
> |:-------:|:-------:|:-------:|:-------:|
> | MATH | 79.29 | 80.00 | 77.86 |
> | MMLU | 72.81 | 78.07 | 74.56 |
>
> | Partial Consensus   | 0.4   | 0.5   | 0.6   |
> |:-------:|:-------:|:-------:|:-------:|
> | MATH | 78.57 | 80.00 | 76.43 |
> | MMLU | 73.68 | 78.07 | 75.44 |
>
> | Conflict Score   | 1   | 2   | 3   |
> |:-------:|:-------:|:-------:|:-------:|
> | MATH | 78.57 | 80.00 | 77.14 |
> | MMLU | 75.44 | 78.07 | 76.32 |
>
> **Q5.** More failure case.
>
> **A5.** Thank you for your suggestion. We add an additional failure case in the table below. This case demonstrates that BCCS is able to correctly identify the incorrect result produced by the agent as unreliable and appropriately selects a conflicting agent as the collaborator. However, due to limitations in model performance, all agents fail to generate the correct answer, thus the agent can not update a correct anwer in the next round.
>
> **Case:**
>
> Question: A cannon is mounted on a truck that moves forward at a speed of 5 m/s. The operator wants to launch a ball from a cannon so the ball goes as far as possible before hitting the level surface. The muzzle velocity of the cannon is 50 m/s. At what angle from the horizontal should the operator point the cannon?  A) 5&deg;, B) 41&deg;, C) 45&deg;, D) 49&deg;
>
> | Opinions | Results |
> |:---------|:---------:|
> | **[Initial Opinion] Agent:** ...the angle that maximizes the range is 45&deg;.| C (Incorrect)|
> | **[Initial Opinion] Conflicting Agent:** The optimal angle from the horizontal for the cannon to achieve the maximum range, considering the truck's speed, is approximately 41 degrees. | B (Incorrect) |
> | **[Update Opinion] Agent:** The provided solution acknowledges that the truck's speed adds more to the horizontal component at lower angles, leading to an optimal angle of 41 degrees. | B (Incorrect) |
>
> **Q6.** Typos correction.
>
> **A6.** We sincerely appreciate your careful reading and valuable feedback. We will correct the typos identified in the final version.

---

> ### Author Response · Authors · 2025-08-04
>
> Thank you for your valuable comments, and we have carefully addressed all of your comments, if there are still any questions requiring further clarification, we sincerely hope you could point them out. If you consider that our revisions have met your expectations, we would be grateful if you could consider raising the evaluation rating accordingly. Your feedback is of utmost importance to us!

---

> > ### Comment · Reviewer_yocy · 2025-08-05
> > **Thanks for the response**
> >
> > I would like to thank the authors for their response. Some of my concerns have been addressed. But I also shared similar concerns with other reviewers about the additional computational cost and algorithm sensitivity. So I decide to maintain my original score.

---

> > > ### Author Response · Authors · 2025-08-06
> > >
> > > Thank you for acknowledging our responses. Regarding computational cost, we have elaborated on how computational cost is assessed, and added the table of comparing computational cost between our proposed BCCS and the SOTA baseline on each dataset **in response A2 in the rebuttal**, showing that our proposed BCCS can achieve better results with lower computational cost. For sensitivity concerns, we have analyzed the sensitivity of the hyperparameters of agent numbers, maximum rounds and leader numbers **in Appendix D.1**, and we have clarified the sensitivity of the thresholds of full consensus, partial consensus and conflict score **in response A4 in the rebuttal**. We hope these empirical results can address your concerns.

---

### Official Review · Reviewer_gdu8 · 2025-07-03

**Clarity:** 3
**Significance:** 2
**Originality:** 3
**Rating:** 4
**Confidence:** 4

**Summary:**

This paper introduces the Belief-Calibrated Consensus Seeking (BCCS) framework, a novel approach to multi-agent consensus seeking for complex Natural Language Processing (NLP) tasks. The BCCS framework addresses limitations in existing methods by considering system-internal beliefs to achieve more stable consensus. It incorporates a belief-calibrated consensus judgment (BCCJ) module, a collaborator assignment (CA) module for partial consensus states, and a leader selection (LS) module for no consensus states, all aimed at guiding agents towards stable and accurate outcomes. The authors provide theoretical foundations for selecting optimal collaborators and leaders and demonstrate the effectiveness of BCCS through experiments on the MATH and MMLU benchmark datasets, where it outperforms current state-of-the-art methods in accuracy.

**Questions:**

- Could the authors elaborate on the specific methodologies used for "Belief Calibration" and how these beliefs are quantitatively measured and updated throughout the collaboration process?

- How does the BCCS framework handle scenarios where no strong leaders emerge or when the identified leaders consistently hold suboptimal beliefs, potentially leading to a "no consensus" state despite iterations?

- What are the computational costs (e.g., time, memory) associated with calculating the "conflict scores" for collaborator assignment, especially as the number of agents and opinion groups increases in more complex scenarios?

**Ethical Concerns:**

["NO or VERY MINOR ethics concerns only"]

**Final Justification:**

The author addresses most of my concerns, and I will increase my rating from 3 to 4.

**Limitations:**

Yes.

**Quality:**

2

**Strengths And Weaknesses:**

## Strengths

- The paper introduces a novel Belief-Calibrated Consensus Judgment (BCCJ) module that not only considers agents' explicit answers but also calibrates them based on their associated belief levels, providing a more robust assessment of consensus compared to traditional voting mechanisms. This addresses the limitation of existing methods that overlook contradictions in system-internal beliefs.

- The framework provides a theoretical basis for selecting optimal collaborators and leaders to maximize consensus stability. The Collaborator Assignment (CA) module intelligently assigns supportive and conflicting collaborators based on uncertainty and belief levels, while the Leader Selection (LS) module identifies high-belief leaders to guide opinion convergence, promoting stable and accurate outcomes.

- The proposed BCCS framework demonstrates superior performance on challenging NLP tasks, outperforming existing multi-agent methods by 2.23% accuracy on MATH and 3.95% accuracy on MMLU benchmark datasets. This suggests its effectiveness in enhancing collective reasoning in multi-agent systems for complex NLP problems.

## Weaknesses

- The approach's performance gains are not consistently significant across all sub-tasks within the benchmark datasets, with greater improvements noted only on "more challenging tasks". This suggests varying effectiveness depending on task complexity.

- The study primarily focuses on mathematical reasoning and natural language understanding tasks. Its generalizability to other complex NLP domains or different types of multi-agent collaboration scenarios is not explored.

- The framework introduces additional computational overhead, as it requires calculating conflict scores for collaborator selection. This could impact its practicality in real-time or resource-constrained environments.

- The optimal number of agents and iteration rounds appear to be empirically determined, with the paper suggesting an "optimal balance" for seven agents and three rounds. This implies a sensitivity to these hyperparameters that might require careful tuning for different applications.

- The analysis of leader impact shows that while higher-belief leaders expedite convergence, those with lower beliefs can reduce consensus reliability. This highlights a potential fragility in the system's robustness if less reliable agents are selected as leaders.

---

> ### Author Rebuttal · Authors · 2025-07-30
>
> We are grateful for your valuable comments. We provide detailed responses to each comment below, aiming to better articulate the contributions of our work. If you have any other questions, please feel free to let us know. Otherwise, if our responses have addressed your comments satisfactorily, we kindly invite you to consider updating the rating.
>
> **Q1.** Significance of performance gains.
>
> **A1.** Thank you for your comment. We agree that the performance gains are more significant on more challenging tasks.
> For simple tasks, since the original model itself can already achieve relatively stable consensus, the marginal benefit of further introducing collaborative mechanisms is relatively limited.
> We will add a clarification of this comment in the final version.
>
> **Q2.** Generalizability to other NLP tasks.
>
> **A2.** Thank you for your suggestion. We extend the evaluation on two widely used NLP benchmark datasets for multi-agent reasoning scenarios, including FOLIO [1] of logical reasoning and CommonsenseQA [2] of commonsense reasoning. We compare our proposed BCCS with three best performed baselines, the results are listed in the table below, which demonstrate that our proposed **BCCS performs better than the baselines on the two datasets**.
>
> |     | FOLIO | CommonsenseQA |
> |:-----:|:-----:|:-----:|
> | CMD | 79.03 | 79.35 |
> | MAD | 80.65 | 78.26 |
> | PARSE | 77.42 | 80.43 |
> | BCCS | **82.26** | **82.61** |
>
> **Q3.** Analysis of computational overhead.
>
> **A3.** We would like to clarify that the conflict score calculation is a lightweight operation that does not require GPU resources, which computation time is practically negligible.
> Therefore, although the number of conflict score calculations increases with the growth of agent numbers and opinion groups, it does not lead to a significant rise in time and space complexity.
> To better understand the computational efficiency, we separately count the average number of tokens per case (denoted by "#Token") as an estimate of computational cost for both our proposed BCCS and the SOTA baseline on each dataset of MATH and MMLU, as shown in the tables below. The results demonstrate that BCCS achieves better results with lower computational cost.
>
> | MATH | BCCS | CMD |
> |:-----:|:-----:|:-----:|
> | #Token | 6554 | 9224 |
>
> | MMLU | BCCS | PARSE |
> |:-----:|:-----:|:-----:|
> | #Token | 2981 | 6349 |
>
> **Q4.** Determination of the hyperparameters of agent numbers and iteration rounds.
>
> **A4.** Thank you for your insightful comment regarding the empirical determination of the optimal number of agents and iteration rounds.
> We would like to clarify that our approach of determination for these hyperparameters is consistent with common practices in the existing multi-agent collaboration systems [3], where the number of agents and interaction rounds are often determined empirically.
>
> **Q5.** Robustness of leader selection in BCCS.
>
> **A5.** Thank you for your valuable comment. We fully agree with the observation that selecting leaders with lower beliefs compromises the system’s robustness by reducing consensus reliability, while higher-belief leaders facilitate faster convergence.
> Our proposed method selects the agent with the highest beliefs as the leaders in each iteration to avoid non-robust outcomes, thereby preventing suboptimal agents from serving as long-term leaders. Situations where all agents have relatively low belief are rare. If such a case does occur, it indicates that none of the agents are capable of solving the problem, making it impossible to accomplish the task through collaboration mechanism, instead it will complete the task through the voting mechanism.
>
> **Q6.** Specific methodologies used for belief calibration.
>
> **A6.** Thank you for your constructive question regarding belief calibration.
> We have already described **the details of the methods of belief calibration in Section 4**. Specifically, the BCCJ module utilizes belief to calibrate the judgment of multi-agent system consensus states based on Byzantine consensus theory.
> When the multi-agent system is in partial consensus state, after the CA module determines the conflict level between opinion groups, it selects the agents with the highest beliefs as the collaborators to enhance the reliability of the multi-agent system.
> When the multi-agent system is in no consensus state, LS module selects the agents with the highest beliefs as the leaders to guide the opinions of other agents.
>
> **References**
>
> [1] Simeng Han, Hailey Schoelkopf, Yilun Zhao, Zhenting Qi, Martin Riddell, Wenfei Zhou and etc. FOLIO: Natural Language Reasoning with First-Order Logic. EMNLP 2024.
>
> [2] Alon Talmor, Jonathan Herzig, Nicholas Lourie, and Jonathan Berant. CommonsenseQA: A Question Answering Challenge Targeting Commonsense Knowledge. NAACL 2019.
>
> [3] Yilun Du, Shuang Li, Antonio Torralba, Joshua B. Tenenbaum, Igor Mordatch. Improving factuality and reasoning in language models through multiagent debate. ICML 2024.

---

> ### Author Response · Authors · 2025-08-04
>
> Thank you for your valuable comments, and we have carefully addressed all of your comments, if there are still any questions requiring further clarification, we sincerely hope you could point them out. If you consider that our revisions have met your expectations, we would be grateful if you could consider raising the evaluation rating accordingly. Your feedback is of utmost importance to us!

---

> > ### Comment · Reviewer_gdu8 · 2025-08-07
> > **Response to author**
> >
> > Thanks for the clarification. The author addresses most of my concerns, and I will increase my rating from 3 to 4.

---

> > > ### Author Response · Authors · 2025-08-08
> > >
> > > We sincerely appreciate the time and effort you have devoted to considering our rebuttal, along with your thoughtful decision to update the rating.

---

### Official Review · Reviewer_p42k · 2025-07-05

**Clarity:** 3
**Significance:** 2
**Originality:** 2
**Rating:** 3
**Confidence:** 2

**Summary:**

This paper introduces the Belief-Calibrated Consensus Seeking (BCCS) framework for multi-agent systems addressing complex NLP tasks.  Motivated by limitations in prior consensus-seeking methods (e.g., over- reliance on voting and indiscriminate agent collaboration), the framework augments consensus evaluation with a novel belief calibration component, optimal collaborator assignment, and leader selection for stable consensus formation.

**Questions:**

1.  Has the study considered how the sensitivity or calibration of generation probability (as a belief signal) varies across different types of LLMs and tasks?
2.  How does BCCS perform in scenarios involving adversarial or noisy agents that may misreport beliefs or answers?
3.  Could the authors provide empirical evidence or ablation studies to verify whether LLM agent behavior and convergence match theoretical predictions?
4. Are there any observed failure modes or limitations of the proposed approach?

**Ethical Concerns:**

["NO or VERY MINOR ethics concerns only"]

**Final Justification:**

I thank the authors for addressing my concerns.Considering the comprehensive score, I am maintaining my score.

**Limitations:**

Yes, the paper provides a brief discussion of limitations in Section 9, noting the lack of evaluation in embodied agents and higher computational cost. However, a more in-depth examination of belief calibration shortcomings, and adversarial agent scenarios would strengthen this section.

**Quality:**

3

**Strengths And Weaknesses:**

Strengths: This paper presents a belief-calibrated multi-agent framework (BCCS) to address the limitations of uncalibrated beliefs and indiscriminate agent interaction in LLM systems.  The proposed solution is supported by both rigorous theoretical analysis and extensive experiments on challenging benchmarks like MATH and MMLU, demonstrating significant accuracy improvements.
Weakness: The evaluation is confined to two reasoning benchmarks (MATH and MMLU), leaving the framework's generalizability to other tasks or adversarial scenarios unclear.    Furthermore, the paper lacks a meaningful analysis of computational scalability, fails to provide a principled motivation for key hyperparameter choices, and does not adequately explore edge cases.

---

> ### Author Rebuttal · Authors · 2025-07-30
>
> We sincerely appreciate your favorable evaluation of the theoretical analysis and empirical results presented in our work, along with your valuable feedback. We have addressed each comment thoroughly below, with the intention of clarifying and further substantiating the significance of our contributions. If you have any other questions, please feel free to let us know. Otherwise, if our responses have addressed your comments satisfactorily, we kindly invite you to consider updating the rating.
>
> **Q1.** Generalizability to other tasks.
>
> **A1.** Thank you for your constructive suggestion. We extend the evaluation on two common used NLP benchmark datasets, including FOLIO [1] of logical reasoning and CommonsenseQA [2] of commonsense reasoning. We compare our proposed BCCS with three best performed baselines, the results are listed in the table below, which demonstrate that our proposed **BCCS performs better than the baselines on the two datasets**.
>
> |     | FOLIO | CommonsenseQA |
> |:-----:|:-----:|:-----:|
> | CMD | 79.03 | 79.35 |
> | MAD | 80.65 | 78.26 |
> | PARSE | 77.42 | 80.43 |
> | BCCS | **82.26** | **82.61** |
>
> **Q2.** Analysis of computational scalability.
>
> **A2.** To better understand the computational scalability, we separately count the average number of tokens per case (denoted by "#Token") as an estimate of computational cost for both our proposed BCCS and the SOTA baseline on each dataset of MATH and MMLU, as shown in the tables below. The results demonstrate that BCCS achieves better performance with lower computational cost.
> Besides, although the conflict score calculation requires additional time, it is a lightweight operation that does not require GPU resources, thus the computation time is practically negligible.
>
> | MATH | BCCS | CMD |
> |:-----:|:-----:|:-----:|
> | #Token | 6554 | 9224 |
>
> | MMLU | BCCS | PARSE |
> |:-----:|:-----:|:-----:|
> | #Token | 2981 | 6349 |
>
> **Q3.** Hyperparameter choices, edge cases and failure modes.
>
> **A3.** The values of hyperparameters including the consensus and conflict thresholds are derived mathematically, with detailed derivations presented in Section 4.
> For other hyperparameters, including agent number, leader number and rounds, we follow the existing multi-agent collaboration methods [3] to select these hyperparameters and provide the comparision of different values of these hyperparameters in Appendix D.1.
> Moreover, Appendix D.3 presents a variety of edge cases, covering both those successfully handled by BCCS and the challenging instance where it fails.
> The dominant failure modes originate from the inherent limitations of the backbone models, which restrict the agents' ability to generate or select correct responses. Our proposed framework aims to mitigate the limitations as much as possible compared to other baselines.
>
> **Q4.** Sensitivity or calibration of generation probability.
>
> **A4.** We sincerely appreciate your insightful question regarding the sensitivity and calibration of generation probability as a belief across different LLMs and tasks. This is an interesting research direction.
> However, our study follows the previous works [4] to estimate belief via probability, we primarily focus on proposing a collaboration framework for improving reasoning performance using beliefs estimated by probabilities, which is a widely used approximation for belief estimation in large language models, rather than enforcing a best configuration of beliefs across all models and tasks.
>
> **Q5.** Performance with misreporting beliefs in adversarial scenarios involving noisy agents.
>
> **A5.** To simulate adversarial and noisy scenarios, we conduct an analysis experiment by misreporting beliefs. Specifically, we introduce adversarial conditions of misreporting beliefs by perturbing the belief of one randomly selected agent in each round (denoted as "AdvNoise"), either by increasing lower beliefs or decreasing higher beliefs. The results are shown in the tables below, which demonstrate that misreporting beliefs can lead to some performance degradation, yet the overall performance remains higher than the SOTA baseline on each dataset. This is because the remaining correct beliefs are still capable of calibrating the inaccurate or misreported answers, demonstrating the robustness of our proposed BCCS in adversarial scenarios involving noisy agents.
>
> | MATH | BCCS | AdvNoise | CMD |
> |:-----:|:-----:|:-----:|:-----:|
> | Accuracy   | 80.00   | 79.29   | 78.57  |
>
> | MMLU | BCCS | AdvNoise | PARSE |
> |:-----:|:-----:|:-----:|:-----:|
> | Accuracy   | 78.07   | 76.32   | 70.18  |
>
> **Q6.** Empirical evidence and ablation studies for matching theoretical predictions.
>
> **A6.** We appreciate your suggestion. We would like to clarify that we have already conducted empirical studies to verify the alignment between LLM agent behavior and theoretical predictions. Specifically:
> * Figures 4 (a)-(b) present the ablation results validating the conclusions in Theorem 3.2, with corresponding analyses located in the first part of Section 6.3.
> * Figures 4 (c)-(d) present the ablation results validating the conclusions in Theorem 3.3, with corresponding analyses located in the second part of Section 6.3.
>
> We hope the analyses in Section 6.3 sufficiently address your concerns.
>
> **References**
>
> [1] Simeng Han, Hailey Schoelkopf, Yilun Zhao, Zhenting Qi, Martin Riddell, Wenfei Zhou and etc. FOLIO: Natural Language Reasoning with First-Order Logic. EMNLP 2024.
>
> [2] Alon Talmor, Jonathan Herzig, Nicholas Lourie, and Jonathan Berant. CommonsenseQA: A Question Answering Challenge Targeting Commonsense Knowledge. NAACL 2019.
>
> [3] Yilun Du, Shuang Li, Antonio Torralba, Joshua B. Tenenbaum, Igor Mordatch. Improving factuality and reasoning in language models through multiagent debate. ICML 2024.
>
> [4] Chiwei Zhu, Benfeng Xu, Quan Wang, Yongdong Zhang, and Zhendong Mao. On the Calibration of Large Language Models and Alignment. Findings of EMNLP 2023.

---

> ### Author Response · Authors · 2025-08-04
>
> Thank you for your valuable comments, and we have carefully addressed all of your comments, if there are still any questions requiring further clarification, we sincerely hope you could point them out. If you consider that our revisions have met your expectations, we would be grateful if you could consider raising the evaluation rating accordingly. Your feedback is of utmost importance to us!

---

> > ### Comment · Reviewer_p42k · 2025-08-06
> >
> > I thank the authors for addressing my concerns. I am maintaining my score.

---

### Note · Authors · 2025-08-16

Dear PCs, SACs, ACs, and Reviewers:

We sincerely appreciate the time and effort you have dedicated to reviewing our paper. Given the extensive reviewer comments and our responses, we have summarized and organized the key points below. We hope this will facilitate your discussion and evaluation of our paper.

**Core contribution:** We provide a theoretical framework for selecting optimal collaborators that maximize consensus stability. Based on the theorems, we propose the Belief-Calibrated Consensus Seeking (BCCS) framework to facilitate stable consensus via selecting optimal collaborators and calibrating the consensus judgment by system-internal beliefs.

**Discussion summary:** The main concerns of all reviewers can be categorized as below:

* Generalizability to other tasks. Reviewer p42k, Reviewer gdu8 and Reviewer npWK questioned the generalizability of our proposed method to other tasks. To clarify these questions, we demonstrated that our proposed method **outperforms the SOTA baselines by 1.61% and 2.18% on the two additional challenging datasets**.
* Analysis of computational cost. All reviewers raised the concerns about the computational cost of our proposed method. To address these concerns, we supplemented the analytical experiments, and elucidated that our proposed method can outperform the SOTA baselines **while reducing generated tokens by at least 28.95%**.
* Sensitivity of hyperparameters. Reviewer yocy and Reviewer npWK raised the concerns about the sensitivity of hyperparameters. To address these concerns, we supplemented the sensitivity analysis experiments on the consensus and conflict thresholds, and for other hyperparameters, including agent number, leader number and rounds, the sensitivity analysis experiments are shown in Appendix D.1.

These concerns mainly involved peripheral or implementation-specific details that do not affect the core methodology or the validity of our results.

**Rebuttal outcome:** By the conclusion of the Author-Reviewer discussion period, Reviewer gdu8 increased the rating from 3 to 4, and most of the reviewers confirmed that their key concerns were addressed.

Sincerely,
The authors

---

### Decision · Program_Chairs · 2025-09-17

**Decision:**

Accept (poster)

**Comment:**

This manuscript investigates the issue of optimally combining the inputs from multiple independent agents to solve complex NLP tasks. The paper introduces the concept of a Belief Calibrated Consensus Seeking system which seeks to optimally combine the opinions of various agents taking into account their contradictions and their internal beliefs.

Strengths:

The reviewers felt that this was an important issue to address

The reviewers appreciated the mathematically derived framework

The reviewers appreciated that the approach did seem to meaningfully improve performance on challenging datasets.

Weaknesses:

Initially concerns were expressed regarding the number of datasets used. This was addressed in the rebuttal phase with the addition of two more challenging datasets that showed performance improvements.

Concerns were raised about the computational effort required by the method which was also addressed to most reviewers satisfaction in the rebuttal.


After the rebuttal and discussion phase 3 of the four reviewers expressed support for the manuscript and were convinced by the arguments advanced by the authors. It appears that the proposed method does provide meaningful improvements in performance across a number of domains as evidenced by the experimental results. While concerns about parameter tuning remained, the reviewers were convinced of the utility of the method.